# LoRA Training Provably Converges to a Low-Rank Global Minimum or It Fails Loudly (But it Probably Won't Fail)

**Junsu Kim** [1]  **Jaeyeon Kim** [2]  **Ernest K. Ryu** [3]

## Abstract

Low-rank adaptation (LoRA) has become a standard approach for fine-tuning large foundation models. However, our theoretical understanding of LoRA remains limited as prior analyses of LoRA's training dynamics either rely on linearization arguments or consider highly simplified setups. In this work, we analyze the LoRA loss landscape without such restrictive assumptions. We define two regimes: a "special regime", which includes idealized setups where linearization arguments hold, and a "generic regime" representing more realistic setups where linearization arguments do not hold. In the generic regime, we show that LoRA training converges to a global minimizer with low rank and small magnitude, or a qualitatively distinct solution with high rank and large magnitude. Finally, we argue that the zero-initialization and weight decay in LoRA training induce an implicit bias toward the low-rank, small-magnitude region of the parameter space—where global minima lie—thus shedding light on why LoRA training usually succeeds in finding global minima.

## 1. Introduction

With the recent explosive trend of scale, fine-tuning a pre-trained foundational model to target downstream tasks has become a dominant approach to deep learning. Low-rank adaptation (LoRA) (Hu et al., 2022) is a parameter-efficient fine-tuning method freezing the pre-trained weight matrix $W_0 \in \mathbb{R}^{m \times n}$, and training a low-rank update $X = AB^\intercal$ to it using

$$W = W_0 + X = W_0 + AB^\intercal,$$

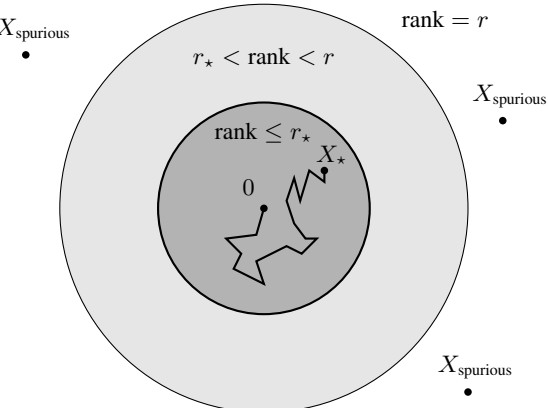

Figure 1: In LoRA fine-tuning, under the assumption that the global minimum $X_\star$ has low rank and small magnitude, we show that spurious local minima $X_{\text{spurious}}$ may exist, but they have high rank and large magnitude.

where $r \ll \min(m, n)$, $A \in \mathbb{R}^{m \times r}$ and $B \in \mathbb{R}^{n \times r}$. The low-rank factor matrices $A$ and $B$ are respectively initialized as a random Gaussian matrix and a zero matrix, leading to $X = 0$ at initialization. By training fewer parameters, LoRA fine-tuning significantly reduces memory usage, making fine-tuning feasible on GPUs with limited GPU memory.

The broad use of LoRA has spurred theoretical works aimed at understanding its effectiveness. One line of work focuses on analyzing LoRA's training dynamics, exploring why optimizers like SGD or Adam successfully find effective low-rank updates despite the significant non-convexity introduced by the factorization $X = AB^\intercal$, as well as the inherent non-convexity of neural networks, by utilizing some degree of linearization. Specifically, Malladi et al. (2023) studies LoRA under a complete linearization, effectively holding $A$ fixed during fine-tuning and viewing the training as a convex optimization problem. A subsequent work (Jang et al., 2024) presents a more refined analysis linearizing with respect to the product $X = AB^\intercal$, retaining the non-convexity arising from the interaction between $A$ and $B$. Beyond linearization, Dayi & Chen (2024) analyzes a two-layer teacher-student setup for rank-1 LoRA. In this work, we carry out a theoretical analysis without any linearizations and any restriction on layers or LoRA rank.

[1]Department of Mathematics, Seoul National University [2]Department of Computer Science, Harvard University [3]Department of Mathematics, University of California, Los Angeles. Correspondence to: Ernest K. Ryu <eryu@math.ucla.edu>.

*Proceedings of the 42^{nd} International Conference on Machine Learning*, Vancouver, Canada. PMLR 267, 2025. Copyright 2025 by the author(s).

**Contribution.** We analyze the loss landscape of LoRA fine-tuning and show that in the "generic regime", a more practical setup where linearization arguments do not hold, a local minimizer is either (i) a global minimizer with small rank and small magnitude or (ii) a spurious local minimizer with high rank and large magnitude. We further argue that the zero-initialization and weight decay in LoRA training induce an implicit bias toward the low-rank, small-magnitude region of the parameter space, where global minima lie. Altogether, we shed light on why practical LoRA training effectively converges to global minima.

Our key assumptions, formally defined and justified in Section 2, are the existence of a low-rank global minimizer for full fine-tuning, restricted strong convexity, and restricted smoothness. Notably, **our analysis does not rely on any linearization arguments**, making it more applicable to practical fine-tuning setups compared to prior work.

## 1.1. Prior works

**PEFT methods and LoRA** Parameter-Efficient Fine-tuning (PEFT) methods have emerged as effective approaches for fine-tuning large language models on downstream tasks while reducing computational and storage requirements. Among numerous proposed methods (Ben Zaken et al., 2022; Li & Liang, 2021; Lester et al., 2021), Low-Rank Adaptation (LoRA) (Hu et al., 2022) has become a predominant approach by decomposing weight updates into low-rank matrices. Several variants such as LoRA+ (Hayou et al., 2024), rsLoRA (Kalajdzievski, 2023), PiSSA (Meng et al., 2024), and MiLoRA (Wang et al., 2025) have been built upon the LoRA framework, addressing the discrepancy with full fine-tuning in optimization and performance.

**Theoretical foundation of LoRA.** Existing theoretical works on LoRA focus on the expressive power and the training dynamics of LoRA. Zeng & Lee (2024) demonstrates that a certain LoRA rank suffices to express a given fine-tuning function. Jang et al. (2024) proves that under the NTK regime, LoRA with rank $\Omega(\sqrt{N})$ can express the global minimizer of the original model. Malladi et al. (2023) argues that the LoRA fine-tuning dynamics are nearly equivalent to the kernel regression. Under this framework, Jang et al. (2024) proves LoRA fine tuning loss has no spurious local minima when the rank is $O(\sqrt{N})$. Beyond the kernel regime, Dayi & Chen (2024) analyzes a two-layer teacher-student setup for LoRA and explains why SGD leads to convergence to a global minimum in this context. Zhang et al. (2025) also identifies the training dynamics in a 2-layer setup, proving LoRA will align to a singular subspace of one-step gradient of full fine-tuning.

**Low-rank optimization.** The low-rank optimization problem

$$\min_{X \in \mathbb{R}^{m \times n}, \, \text{rank}(X) \leq r} f(X)$$

has been extensively studied in the optimization literature, including matrix sensing (Recht et al., 2010) and matrix completion (Candès & Recht, 2012). Rather than directly optimizing over the space of low-rank matrices, it is often preferred to employ the Burer-Monteiro factorization (Burer & Monteiro, 2003), which formulates the problem by parameterizing $X$ as $X = UV^{\mathsf{T}}$, $U \in \mathbb{R}^{m \times r}$, $V \in \mathbb{R}^{n \times r}$.

As the Burer-Monteiro factorization introduces nonconvexity, a large body of work has identified conditions under which this approach avoids spurious local minima (Bhojanapalli et al., 2016; Ge et al., 2017; Park et al., 2017; Zhang, 2021). Further studies extend these results to general settings (Ha et al., 2020; Zhang, 2024). In our work, we utilize the framework established in these studies with novel techniques to extend its boundary to optimization guarantees in LoRA training.

## 1.2. Notation and preliminaries

**Matrix notation.** For $X \in \mathbb{R}^{m \times n}$, denote its singular values as $\sigma_1(X) \geq \sigma_2(X) \geq \cdots \geq \sigma_r(X) \geq 0$. For matrices $A$ and $B$, let $\|A\|_2 = \sigma_1(A)$ denote the spectral norm, $\|A\|_* = \sum \sigma_i(A)$ the nuclear norm, $\|A\|_F = \sqrt{\sum \sigma_i(A)^2}$ the Frobenius norm, and $\langle A, B \rangle = \text{tr}(A^{\mathsf{T}}B)$ the matrix inner product. For a tuple of matrices $\mathbf{A} = (A^{(1)}, \ldots, A^{(L)})$, denote $\|\mathbf{A}\| = \sum_{l=1}^{L} \|A^{(l)}\|$ for any matrix norm $\|\cdot\|$ and $\text{rank}(\mathbf{A}) = \max_{1 \leq l \leq L} \text{rank}(A^{(l)})$.

**Neural network.** Let $f(\,\cdot\,;\cdot) : \mathcal{P} \times \mathcal{X} \to \mathbb{R}^K$ be a neural network where $\mathcal{P}$ is the parameter space, $\mathcal{X}$ is the data space, and $\mathbb{R}^K$ is the output space. Assume the model is pre-trained to $\Theta_0 \in \mathcal{P}$, i.e., the pre-trained model is $f(\Theta_0;\cdot)$.

**Fine-tuning loss.** Let $\mathbf{W}_0 = (W_0^{(1)}, \ldots, W_0^{(L)}) \subset \Theta_0$ be the pre-trained value of the weights $\mathbf{W}$ that we choose to fine-tune. We wish to fine-tune the pre-trained model $f(\Theta_0;\cdot)$ on a downstream task with data distribution $(x, y) \sim \mathcal{D}$. With slight abuse of notation, write $f(\mathbf{W};\cdot)$ to denote $f(\Theta;\cdot)$, where all parameters of $\Theta$ excluding $\mathbf{W}$ are fixed to their corresponding values in $\Theta_0$. Let

$$\mathbf{X} = (X^{(1)}, \ldots, X^{(L)})$$

be the change of $\mathbf{W}$ during the (full) fine-tuning. The true objective one hopes to minimize is

$$\mathcal{L}^{\text{full}}(\mathbf{X}) = \mathbb{E}_{(x,y) \sim \mathcal{D}} \left[ \ell(f(\mathbf{W}_0 + \mathbf{X}; x), y) \right]$$

with some loss function $\ell(\cdot, \cdot)$. We assume $\ell(x, y)$ is nonnegative and twice-differentiable with respect to $x$ for any $y$. In practice, we have access to a finite dataset $\{(x_i, y_i)\}_{i=1}^{N}$,

so we minimize the empirical risk

$$\widehat{\mathcal{L}}^{\text{full}}(\mathbf{X}) = \frac{1}{N} \sum_{i=1}^{N} \ell(f(\mathbf{W}_0 + X; x_i), y_i).$$

**LoRA.** Low-rank adaptation (LoRA) uses a rank-$r$ parameterization for each update matrix

$$X^{(l)} = A^{(l)}(B^{(l)})^{\intercal} \in \mathbb{R}^{m_l \times n_l}$$

with $A^{(l)} \in \mathbb{R}^{m_l \times r}$ and $B^{(l)} \in \mathbb{R}^{n_l \times r}$ for $l = 1, \ldots, L$. Denote

$$\mathbf{A} = (A^{(1)}, \ldots, A^{(L)}), \quad \mathbf{B} = (B^{(1)}, \ldots, B^{(L)})$$

and

$$\mathbf{A}\mathbf{B}^{\intercal} = \left( A^{(1)}(B^{(1)})^{\intercal}, \ldots, A^{(L)}(B^{(L)})^{\intercal} \right).$$

Under this parametrization, we define the empirical LoRA risk as

$$\widehat{\mathcal{L}}^{\text{lora}}(\mathbf{A}, \mathbf{B}) \triangleq \widehat{\mathcal{L}}^{\text{full}}(\mathbf{A}\mathbf{B}^{\intercal})$$

We adopt the standard initialization (Hu et al., 2022), respectively initializing each $\mathbf{A}$ and $\mathbf{B}$ as a random gaussian and zero, leading to $\mathbf{A}\mathbf{B}^{\intercal} = 0$ at initialization.

**Second-order stationary points.** Let $L: \mathbb{R}^n \to \mathbb{R}$ be twice-continuously differentiable. We say $X \in \mathbb{R}^n$ is a *(first-order) stationary point* if

$$\nabla L(X) = 0.$$

We say $X \in \mathbb{R}^n$ is a *second-order stationary point (SOSP)* if

$$\nabla L(X) = 0, \quad \nabla^2 L(X)[U, U] \geq 0,$$

for any $U \in \mathbb{R}^n$. Lastly, we say $X \in \mathbb{R}^n$ is a *local minimum* if there exists an open ball $\mathcal{B}$ that contains $X$ and

$$L(X) \leq L(X')$$

for any $X' \in \mathcal{B}$. It follows that a local minimum is an SOSP. If a local minimum is not a global minimum, we say it is a *spurious* local minimum.

Prior works have established that stochastic gradient descent applied to twice-continuously differentiable functions (regardless of convexity) roughly converges to SOSPs.

**Theorem** (Theorem 4.1 of Lee et al. (2016)). *Gradient descent on twice-differentiable functions with random initialization, almost surely, does not converge to strict saddle points. I.e.,if gradient descent converges, it converges to an SOSP, almost surely.*

**Theorem** (Informal, Theorem 1 of Ge et al. (2015)). *Stochastic gradient descent with noise on twice-differentiable strict saddle functions (i.e., every stationary point is either a local minimum or a strict saddle) does not converge to strict saddle points with high probability. I.e., if stochastic gradient descent with noise converges, it converges to an SOSP with high probability.*

In the context of our work, the implication is that LoRA training converges to SOSPs. The question we address is whether such SOSPs are global minima or whether it is possible to converge to a bad local minimum.

### 1.3. Weight decay and nuclear norm regularization

Let $\lambda \geq 0$ and

$$\widehat{\mathcal{L}}^{\text{lora}}_{\lambda}(\mathbf{A}, \mathbf{B}) \triangleq \widehat{\mathcal{L}}^{\text{lora}}(\mathbf{A}, \mathbf{B}) + \frac{\lambda}{2} \left( \|\mathbf{A}\|_F^2 + \|\mathbf{B}\|_F^2 \right).$$

Practical LoRA training typically employs weight decay (Hu et al., 2022; Dettmers et al., 2023) and applying SGD with weight decay on $\widehat{\mathcal{L}}^{\text{lora}}$ is equivalent to minimizing $\widehat{\mathcal{L}}^{\text{lora}}_{\lambda}$ without weight decay. In other words, the effect of weight decay is equivalent to adding $\ell_2$-regaularization. Let

$$\widehat{\mathcal{L}}^{\text{full}}_{\lambda}(\mathbf{X}) \triangleq \widehat{\mathcal{L}}^{\text{full}}(\mathbf{X}) + \lambda \|\mathbf{X}\|_{\star}.$$

From the prior literature on low-rank matrix sensing and Burer–Monterio factorizations (Recht et al., 2010, Lemma 5.1), it is known that minimizing this $\ell_2$-regularized problem in $\mathbf{A}$ and $\mathbf{B}$ is mathematically equivalent to minimizing the nuclear-norm regularized loss in the product $\mathbf{X} = \mathbf{A}\mathbf{B}^{\intercal}$ subject to a rank constraint. In other words

$$\underset{\mathbf{A}, \mathbf{B}}{\text{minimize}} \quad \widehat{\mathcal{L}}^{\text{lora}}_{\lambda}(\mathbf{A}, \mathbf{B}) \quad \Leftrightarrow \quad \begin{array}{c} \underset{\mathbf{X}}{\text{minimize}} \quad \widehat{\mathcal{L}}^{\text{full}}_{\lambda}(\mathbf{X}) \\ \text{subject to} \quad \text{rank}(\mathbf{X}) \leq r \end{array}$$

When using LoRA, we hope to match the performance of full fine-tuning. We expect this to be feasible if the full fine-tuning problem (with nuclear norm regularization) admits a global minimizer whose rank is at most $r$, since the LoRA update $\mathbf{A}\mathbf{B}^{\intercal}$ cannot represent updates of rank larger than $r$. Therefore, as we discuss further in Section 2.1, we conduct our analysis under the assumption that $\widehat{\mathcal{L}}^{\text{full}}_{\lambda}$ has a low-rank global minimizer.

**Nuclear norm regularization.** Nuclear norm regularization is a popular technique that promotes low-rank solutions in matrix optimization. As the convex envelope of the rank function on the unit ball (Fazel et al., 2001), the nuclear norm penalty provides a tractable alternative to directly minimizing rank. Its effectiveness in yielding low-rank solutions has been demonstrated both theoretically and empirically across various fields, including matrix sensing (Recht et al., 2010), computer vision (Cabral et al., 2013), nonconvex

Table 1: Global minimizer rank $\text{rank}(X_\star)$ as a function of weight decay value $\lambda$.

| SST2 | Max Rank | 749 | 107 | 5 | 3 | 1 |
|------|----------|-----|-----|---|---|---|
| | $\lambda$ | 0.0 | 0.001 | 0.005 | 0.01 | 0.1 |
| CIFAR100 | Max Rank | 752 | 23 | 12 | 4 | 1 |
| | $\lambda$ | 0.0 | 0.0005 | 0.001 | 0.003 | 0.005 |

optimization (Hu et al., 2021), deep learning (Kobayashi et al., 2024), and LoRA (Jang et al., 2024). Collectively, these prior results make the assumption that $\widehat{\mathcal{L}}_\lambda^{\text{full}}$ admits a low-rank minimizer more natural.

## 2. Main assumptions

In this section, we define and quickly justify the main assumptions used in our analyses of Section 3.

### 2.1. Existence of a low-rank minimizer

Throughout our analysis, we assume that **there exists a rank $r_\star$ global minimizer of full fine-tuning loss $\widehat{\mathcal{L}}_\lambda^{\text{full}}$** and that **our LoRA module uses rank $r \geq r_\star$**.

We argue that there is sufficient conceptual and experimental justification supporting the assumption. Initially, LoRA (Hu et al., 2022) was proposed based on the insight that learned over-parameterized models lie in a low intrinsic dimension (Li et al., 2018; Aghajanyan et al., 2021), making them amenable to low-rank updates during fine-tuning. Moreover, as discussed in Section 1.3, training LoRA with weight decay is equivalent to nuclear norm regularization in full fine-tuning, thereby strongly biasing the solution toward low rank. As shown in Table 1 and further discussed in Section 4, we experimentally verify the low-rank assumption in a few setups. Finally, the extensive empirical literature demonstrating the success of LoRA with small rank $r$ further justifies this assumption.

Nevertheless, it may sometimes be more realistic to assume that the global minimizer of full fine-tuning is only *approximately* low rank. We address this issue in Section 3.3, where we generalize the analysis to the case where the global minimizer of full fine-tuning is not exactly low rank.

### 2.2. Restricted strong convexity and smoothness

Our analyses also rely on the assumptions of restricted smoothness and restricted strong convexity, which are weaker assumptions compared to the smoothness and strong convexity assumptions commonly used in optimization.

We say a twice-differentiable function $f \colon \mathbb{R}^{m \times n} \to \mathbb{R}$ is $(\alpha, r, D)$-**restricted strongly convex** about $X_\star$ if

$$\langle \nabla f(X) - \nabla f(X_\star), X - X_\star \rangle \geq \alpha \|X - X_\star\|_F^2.$$

for any $X \in \mathbb{R}^{m \times n}$ such that $\|X - X_\star\|_F \leq D$ and $\text{rank}(X) \leq r$. We denote the largest $\alpha$ such that $f$ is $(\alpha, r, D)$-restricted strongly convex about $X_\star$ as the $(r, D)$-**RSC constant** of $f$ about $X_\star$.

We say a twice-differentiable function $f \colon \mathbb{R}^{m \times n} \to \mathbb{R}$ is $(\beta, r, D)$-**restricted smooth** about $X_\star$ if

$$\nabla^2 f(X)[UX + XV, UX + XV] \leq \beta \|UX + XV\|_F^2$$

for any $[X \in \mathbb{R}^{m \times n}$ such that $\|X - X_\star\|_F \leq D$ and $\text{rank}(X) \leq r]$, $[U \in \mathbb{R}^{m \times m}$ such that $\|U\|_F = \|V\|_F = 1]$ and $\text{rank}(U) = 1]$, and $[V \in \mathbb{R}^{n \times n}$ such that $\|V\|_F = 1$ and $\text{rank}(U) = \text{rank}(V) = 1]$. We denote the smallest $\beta$ such that $f$ is $(\beta, r, D)$-restricted smooth about $X_\star$ (or $\beta = \infty$ if there is no such finite value) as the $(r, D)$-**RSM constant** of $f$ about $X_\star$.

In this work, we consider the case where $\alpha > 0$ and $\beta < \infty$. Although deep learning objectives are typically neither strongly convex nor have small smoothness constants, the *restricted* notions of strong convexity and smoothness are valid in many practical fine-tuning scenarios as we empirically demonstrate in Section 4. Finally, this current definition treats $f$ as a function of a single matrix $X$. In Section 3.2, we generalize the definitions to $\mathbf{X} = (X^{(1)}, X^{(2)}, \ldots, X^{(L)})$ with multiple matrices.

## 3. Spurious Local minima of LoRA

In this section, we analyze the loss landscape of LoRA fine-tuning and show that in the "generic regime", a second-order stationary point (SOSP) is either (i) a global minimizer with small rank and small magnitude or (ii) a spurious solution with high rank and large magnitude.

Section 3.1 starts by presenting the result in the simpler setup of fine-tuning a single matrix when a low-rank global minimizer exists. Section 3.2 extends the result to the setup of fine-tuning multiple matrices. Section 3.3 extends the theory to work when an *approximately* low-rank global minimizer exists. The extensions of Sections 3.2 and 3.3 slightly complicate the notation, but the qualitative conclusion is maintained. In Section 3.4, we discuss why first-order optimizers with zero-initialization and weight decay, are unlikely to converge to the spurious local minimizers.

### 3.1. LoRA converges to a global minimizer or fails loudly

We now state our main result.

**Theorem 1.** *Let $\lambda \geq 0$. Assume the full fine-tuning loss $\widehat{\mathcal{L}}_\lambda^{\text{full}}$ has a rank-$r_\star$ global minimizer $X_\star$. Respectively denote the $(r, D)$-RSC and $(r, D)$-RSM constants of $\widehat{\mathcal{L}}^{\text{full}}$ about $X_\star$ as $\alpha$ and $\beta$. Assume $\alpha > 0$ and $\beta < \infty$. Assume we use a LoRA module with rank $r \geq r_\star$. Then, every SOSP $(A, B)$ of $\widehat{\mathcal{L}}_\lambda^{\text{lora}}$ with $X_\square = AB^\intercal$ and $\|X_\square - X_\star\|_F \leq D$*

*satisfies the following.*

1. *If $2\alpha > \beta$ (special regime), $X_\square$ is a global minimum.*

2. *If $2\alpha \leq \beta$ (generic regime), one of the following holds.*

   (i) *$X_\square$ is a global minimum.*

   (ii) *$X_\square$ is not a global minimum, $\mathrm{rank}(X_\square) = r$ with $\sigma_r(X_\square) \geq \frac{2\alpha}{\beta}\sigma_{r_\star}(X_\square)$, and*

$$\|X_\square - X_\star\|_F^2 \geq \frac{\|X_\square - \Pi_{\mathrm{rank} \leq r_\star}(X_\square)\|_F^2}{1 - \frac{2\alpha\sigma_{r_\star}}{\beta\sigma_r}},$$

   *where $\Pi_{\mathrm{rank} \leq r_\star}(X_\square)$ is the projection of $X_\square$ onto the set of matrices of rank $r_\star$ or less.*

*To clarify, when we say $X_\square$ is or is not a global minimum, it is with respect to $\widehat{\mathcal{L}}_\lambda^{\mathrm{full}}$.*

We denote $2\alpha > \beta$ as the *special regime*, as the loss objective should be very well-conditioned to fall in this regime. Most practical setups would fall into the *generic regime* with $\beta \geq 2\alpha$, thereby being the regime of primary interest.

The global minimizer $X_\star$ of the full fine-tuning loss $\widehat{\mathcal{L}}_\lambda^{\mathrm{full}}$ is assumed to be low rank, and we intuitively understand that $X_\star$ should have small magnitude since we are fine-tuning. Theorem 1 states that in the generic regime, there may be additional spurious local minima, but those will have high rank and will be far away from the global minimizer $X_\star$.

The following corollary restates Theorem 1 in an alternate form that clarifies its main conclusions.

**Corollary 1.** *Consider the setup in Theorem 1. Further assume the strict inequality $r > r_\star$. Let $(A, B)$ be a SOSP of $\widehat{\mathcal{L}}_\lambda^{\mathrm{lora}}$ with $X_\square = AB^\mathsf{T}$ and $\|X_\square - X_\star\|_F \leq D$. Then,*

(i) *If $\sigma_r(X_\square) \leq \frac{2\alpha}{\beta}\sigma_{r_\star}(X_\square)$, then $X_\square$ is a global minimizer.*

(ii) *If $\sigma_r(X_\square) > \frac{2\alpha}{\beta}\sigma_{r_\star}(X_\square)$, then $X_\square$ is a spurious solution, and further $X_\square$ has large magnitude with*

$$\|X_\square\|_F \geq \sqrt{\frac{\sum_{s=r_\star+1}^{r} \sigma_s^2(X_\square)}{1 - \frac{2\alpha\sigma_{r_\star}}{\beta\sigma_r}}} - \|X_\star\|_F.$$

**LoRA training converges to a global minimizer or fails loudly.** As discussed in Section 1.2, Lee et al. (2016) and Ge et al. (2015) imply that LoRA fine-tuning with SGD converges to a SOSP. In Section 3.4, we argue why it is likely that the SOSP we converge to is a global minimizer.

However, if LoRA fine-tuning does converge to a spurious solution, its high rank and large magnitude would be

noticeable, and, as the experiments in Section 4 show, generalization will be poor. In this sense, we describe this mode of failure to be "failing loudly."

**Relation to prior work.** Interestingly, Theorem 1 completely includes the prior loss landscape analysis of (Jang et al., 2024), which considers a linearized loss in the NTK regime with an $\varepsilon$-perturbation. This perturbation ensures $2\alpha = 2\varepsilon > \beta = \varepsilon$, placing the loss objective in the special regime. Then, with Theorem 1, we conclude that any SOSP is a global minimum.

### 3.1.1. PROOF OUTLINE OF THEOREM 1

Our proof technique takes inspiration from the low-rank optimization literature. In fact, the analysis in the special regime $2\alpha \geq \beta$ naturally extends results from matrix sensing (Zhu et al., 2018; Ha et al., 2020). On the other hand, the analysis on the generic regime $2\alpha < \beta$ is a novel result of ours. In the matrix sensing setting, showing that local minimizers near the solution are global minimizers has limited meaning since there is not a good estimate of the global minimizer, so such results were not pursued. On the other hand, in the LoRA fine-tuning setup, 0, the pre-trained baseline, is a good estimate of the global minimizer.

We defer the full proof of Theorem 1 to Appendix A, providing a brief outline here. For notational simplicity, write $f(X) = \widehat{\mathcal{L}}^{\mathrm{full}}(X)$ and $g(A, B) = \widehat{\mathcal{L}}^{\mathrm{lora}}(A, B)$, and $X = X_\square$. (So $X$ is assumed to be an SOSP.) Denote the compact SVD of $X$ as $L_X \Sigma_X R_X^\mathsf{T}$, and $\sigma_i, u_i, v_i$ as the $i$-th (largest) singular value of $X$ and the corresponding singular vectors. From the first and second-order optimality of $g(A, B)$, we acquire the following properties:

1. $0 = \nabla_A g(A, B) = \nabla f(X) \cdot B + \lambda A$

2. $0 = \nabla_B g(A, B) = \nabla f(X)^\mathsf{T} \cdot A + \lambda B$

3. $\nabla^2 g(A, B)[(U, V), (U, V)] = 2\langle \nabla f(X), UV^\mathsf{T} \rangle + \nabla^2 f(X)[AV^\mathsf{T} + UB^\mathsf{T}, AV^\mathsf{T} + UB^\mathsf{T}] + \lambda(\|U\|_F^2 + \|V\|_F^2) \geq 0$ for any $(U, V)$.

Properties 1 and 2 imply $\nabla f(X)$ can be represented as

$$\nabla f(X) = -\lambda L_X R_X^\mathsf{T} + S, \quad L_X^\mathsf{T} S = S R_X = 0 \quad (1)$$

for some matrix $S$. Furthermore, plugging in $(U, V) = (-u_\star u_r^\mathsf{T} A, v_\star v_r^\mathsf{T} B)$ into property 3 and using the $\beta$-restricted smoothness of $f$, where $(u_\star, v_\star)$ are the top singular vectors of $S$, we find

$$\|S\|_2 \leq \lambda + \beta\sigma_r. \quad (2)$$

From (1), (2) we can induce there exists a subgradient $g \in \partial(\lambda\|X\|_*)$ such that $\|g + \nabla f(X)\|_2 \leq \beta\sigma_r$.

Denoting $Z = g + \nabla f(X)$ and $\kappa = \frac{\sigma_{r_\star}}{\beta \sigma_r}$, we see $\|\kappa Z\| \leq \sigma_{r_\star}$ and therefore the top $r_\star$ singular vectors of $X - \kappa Z$ coincide with those of $X$. Thus, by the Eckart–Young–Mirsky Theorem, we have

$$\Pi_{\mathrm{rank} \leq r_\star}(X) \in \underset{\mathrm{rank}(Y) \leq r_\star}{\mathrm{argmin}} \|Y - (X - \kappa Z)\|_F^2.$$

Since $\mathrm{rank}(X_\star) = r_\star$,

$$\|X^{r_\star} - X + \kappa Z\|_F^2 \leq \kappa X \|X_\star - X + \kappa Z\|_F^2$$

which is again equivalent to

$$\|X^{r_\star} - X\|_F^2 \leq \|X_\star - X\|_F^2 + 2\kappa\langle X_\star - X, Z\rangle$$

Now $\alpha$-restricted convexity at $X_\star$ implies

$$\langle X - X_\star, \nabla f(X) - \nabla f(X_\star)\rangle \geq \alpha \|X - X_\star\|_F^2$$

By the global optimality of $X_\star$, from Mordukhovich & Shao (1995, Theorem 3.1) we have $-\nabla f(X_\star) \in \partial(\lambda\|X_\star\|_*)$ and thus the subgradient property implies

$$\langle X_\star, g - (-\nabla f(X_\star))\rangle \geq 0$$

Summing up the three inequalities, we have

$$(2\kappa\alpha - 1)\|X_\star - X\|_F^2 + \|X^{r_\star} - X\|_F^2 \leq 0$$

Therefore when $2\kappa\alpha > 1$, $X_\star = X$, and when $2\kappa\alpha < 1$ the inequality of the theorem holds. $\qquad\square$

### 3.2. Extension to fine-tuning multiple matrices

For the sake of notational convenience, Theorem 1 was stated for the case of fine-tuning a single weight matrix. In this section, we generalize the result to the case of fine-tuning multiple matrices.

First, we extend the definition of restricted smoothness and strong convexity to the multiple matrix case.

Let $f \colon \mathbb{R}^{m_1 \times n_1} \times \cdots \times \mathbb{R}^{m_L \times n_L} \to \mathbb{R}$ be twice differentiable. Let $\mathbf{X} = (X^{(1)}, X^{(2)}, \ldots, X^{(L)})$, $\alpha = (\alpha^{(1)}, \ldots, \alpha^{(L)})$, and $\beta = (\beta^{(1)}, \ldots, \beta^{(L)})$.

We say $f$ is $(\alpha, r, D)$-restricted strongly convex about $X_\star$ if for each $1 \leq l \leq L$,

$$\langle \nabla_l f(\mathbf{X}_\star) - \nabla_l f(\mathbf{X}), X^{(l)} - X_\star^{(l)}\rangle \geq \alpha^{(l)}\|X^{(l)} - X_\star^{(l)}\|_F^2.$$

for any $\mathbf{X}$ such that $\|\mathbf{X} - \mathbf{X}_\star\|_F \leq D$. We denote the tuple $\alpha$ of the largest $\alpha^{(l)}$s such that $f$ is $(\alpha, r, D)$-restricted strongly convex about $\mathbf{X}_\star$ as the $(r, D)$-RSC constant of $f$ about $\mathbf{X}_\star$.

We say a twice-differentiable function $f \colon \mathbb{R}^{m \times n} \to \mathbb{R}$ is $(\beta, r, D)$-restricted smooth about $\mathbf{X}_\star$ if for each $1 \leq l \leq L$,

$$\nabla_{l,l}^2 f(\mathbf{X})[UX^{(l)} + X^{(l)}V, UX^{(l)} + X^{(l)}V]$$
$$\leq \beta^{(l)}\|UX^{(l)} + X^{(l)}V\|_F^2$$

for any $\mathbf{X}$ such that $\|\mathbf{X} - \mathbf{X}_\star\|_F \leq D$, $U \in \mathbb{R}^{m_l \times m_l}$ such that $\mathrm{rank}(U) = 1$ and $\|U\|_F = 1$, $V \in \mathbb{R}^{n_l \times n_l}$ such that $\mathrm{rank}(V) = 1$ and $\|V\|_F = 1$. We denote the tuple $\beta$ of the largest $\beta^{(l)}$ such that $f$ is $(\beta, r, D)$-restricted strongly convex about $X_\star$ as the $(r, D)$-RSM constant of $f$ about $\mathbf{X}_\star$. Here $\nabla_l, \nabla_{l,l}^2$ refers to the gradient and Hessian respect to the $l$th matrix $X^{(l)}$.

Next, under this extended notion of restricted smoothness and convexity, we present the natural extension of Theorem 1 below. The proof follows the same reasoning as in Theorem 1 and is detailed in Appendix A

**Theorem 2.** *Let* $\lambda \geq 0$. *Assume the full fine-tuning loss* $\widehat{\mathcal{L}}_\lambda^{\mathrm{full}}$ *has a rank-$r_\star$ global minimizer* $\mathbf{X}_\star = (X_\star^{(1)}, \ldots, X_\star^{(L)})$. *Respectively denote the* $(r, D)$-RSC *and* $(r, D)$-RSM *constants of* $\widehat{\mathcal{L}}^{\mathrm{full}}$ *about* $\mathbf{X}_\star$ *as* $\alpha = (\alpha^{(1)}, \ldots, \alpha^{(L)})$ *and* $\beta = (\beta^{(1)}, \ldots, \beta^{(L)})$. *Assume* $\alpha^{(1)}, \ldots, \alpha^{(L)} > 0$ *and* $\beta^{(1)}, \ldots, \beta^{(L)} < \infty$. *Assume we use LoRA modules all with rank* $r \geq r_\star$. *Then, every SOSP* $(\mathbf{A}, \mathbf{B})$ *of* $\widehat{\mathcal{L}}_\lambda$ *with* $\mathbf{X}_\square = \mathbf{A}\mathbf{B}^\intercal$ *and* $\|\mathbf{X}_\square - \mathbf{X}_\star\|_F \leq D$ *satisfies the following.*

1. *If* $2\alpha^{(l)} \geq \beta^{(l)}$ *for all* $l = 1, \ldots, L$ *(special regime),* $\mathbf{X}_\square$ *is a global minimum*

2. *If* $2\alpha^{(l)} < \beta^{(l)}$ *for some* $l = 1, \ldots, L$ *(generic regime), one of the following holds.*

   (i) $\mathbf{X}_\square$ *is a global minimum.*

   (ii) $\mathbf{X}_\square$ *is not a global minimum,* $X_\square^{(l)}$ *is exactly rank* $r$ *with* $\sigma_r(X_\square^{(l)}) > \frac{2\alpha^{(l)}}{\beta^{(l)}}\sigma_{r_\star}(X_\square^{(l)})$ *and*

$$\left\|X_\square^{(l)} - X_\star^{(l)}\right\|_F^2 \geq \frac{\left\|X_\square^{(l)} - \Pi_{\mathrm{rank} \leq r_\star}(X_\square^{(l)})\right\|_F^2}{1 - \frac{2\alpha^{(l)}\sigma_{r_\star}}{\beta^{(l)}\sigma_r}}$$

   *for some* $l = 1, \ldots, L$, *where* $\Pi_{\mathrm{rank} \leq r_\star}(X_\square^{(l)})$ *is the projection of* $X_\square^{(l)}$ *onto the set of matrices of rank* $r_\star$ *or less.*

*To clarify, when we say* $\mathbf{X}_\square$ *is or is not a global minimum, it is with respect to* $\widehat{\mathcal{L}}_\lambda^{\mathrm{full}}$.

### 3.3. Extension to approximately low-rank minimizers

In Sections 3.1 and 3.2, we assumed the nuclear-norm regularized full fine-tuning loss $\widehat{\mathcal{L}}_\lambda^{\mathrm{full}}$ has a low-rank minimizer, but this assumption may be unrealistic especially when the weight-decay parameter $\lambda$ is too small. In this section, we relax this assumption and consider the case where $\widehat{\mathcal{L}}_\lambda^{\mathrm{full}}$ has an *approximately* low-rank minimizer. As in Section 3.1, we present here the result for the single matrix case. In Appendix A, we provide a 'Master Theorem' that combines the generalizations of Theorems 2 and 3.

We say $X_\star^{(\delta)}$ is a *$\delta$-global minimizer of full fine-tuning* if

$$\|X_\star^{(\delta)} - X_\star\|_F \leq \delta$$

for some $X_\star$ that exactly minimizes $\widehat{\mathcal{L}}^{\text{full}}$.

**Theorem 3.** *Let $\varepsilon > 0$ and $\lambda \geq 0$. Assume the full fine-tuning loss $\widehat{\mathcal{L}}_\lambda^{\text{full}}$ has a rank-$r_\star$ $\delta$-global minimizer $X_\star^{(\delta)}$ with $\delta = o(\varepsilon^3)$. Respectively denote the $(r, D)$-RSC and $(r, D)$-RSM constants of $\widehat{\mathcal{L}}^{\text{full}}$ about $X_\star$ as $\alpha$ and $\beta$. Assume $0 < \alpha$ and $\beta < \infty$. Assume we use a LoRA module with rank $r \geq r_\star$. Then, every SOSP $(A, B)$ of $\widehat{\mathcal{L}}_\lambda^{\text{lora}}$ with $X_\square = AB^\intercal$ and $\|X_\square - X_\star\|_F \leq D$ satisfies the following.*

1. *If $2\alpha \geq \beta(1+\varepsilon)$ (special regime), $X_\square$ is an $\varepsilon$-global minimizer.*

2. *If $2\alpha < \beta(1+\varepsilon)$ (generic regime), one of the following holds.*

   (i) *$X_\square$ is an $\varepsilon$-global minimizer.*

   (ii) *$X_\square$ is not an $\varepsilon$-global minimizer, $X_\square$ is exactly rank $r$ with $\sigma_r(X_\square) \geq \max\{\frac{2\alpha}{\beta(1+\varepsilon)}\sigma_{r_\star}(X_\square), \frac{\alpha}{2\beta\sqrt{r}}\cdot\varepsilon\}$, and either*

   $$\sigma_r(X_\square) \leq \frac{2\alpha}{\beta}\sigma_{r_\star}(X_\square)$$

   *or*

   $$\|X_\square - X_\star\|_F \geq \sqrt{\frac{\left\|X_\square - \Pi_{\text{rank}\leq r_\star}(X_\square)\right\|_F^2 - \varepsilon^3}{1 - \frac{2\alpha\sigma_{r_\star}}{\beta\sigma_r}}} - \varepsilon^2$$

   *where $\Pi_{\text{rank}\leq r_\star}(X_\square)$ is the projection of $X_\square$ onto the set of matrices of rank $r_\star$ or less.*

*To clarify, when we say $X_\square$ is or is not an $\varepsilon$-global minimizer, it is with respect to $\widetilde{\mathcal{L}}_\lambda^{\text{full}}$.*

### 3.4. LoRA training probably won't fail; it probably won't converge to spurious local minima

In the analysis of Section 3.1 and its subsequent generalizations, we showed that in the generic regime, spurious local minima may exist, but if the training converges to them, this will be very noticeable (failing loudly), as the spurious solutions have high rank and large magnitude. In this section, we argue that the standard LoRA fine-tuning procedure induces implicit biases that make it unlikely for the LoRA training to converge to these spurious local minima.

**Zero-initialization biases the optimization towards minima with smaller magnitude.** LoRA fine-tuning is initialized with $B = 0$, leading to $X = AB^\intercal = 0$ at initialization. This choice comes from the intuition that fine-tuning should not change the model too much, i.e., that $X_\star$ should be small, so the initialization should be at $0$.

When weight decay is used, we can make this argument further quantitative. The global minimizer $X_\star$ satisfies

$$\widehat{\mathcal{L}}(X_\star) + \lambda\|X_\star\|_* \leq \widehat{\mathcal{L}}(0) + \lambda\|0\|_*,$$

thus $\|X\|_* < \frac{\widehat{L}(0)}{\lambda}$. Here, $\widehat{\mathcal{L}}(0)$ is the loss corresponding to directly applying the pre-trained model to the fine-tuning task, so $\widehat{\mathcal{L}}(0)$ should not be inordinately large when the fine-tuning task is not too different from tasks seen during pre-training.

On the other hand, spurious local minima exist only outside a neighborhood of zero, as argued in Corollary 1. Because the SGD or Adam optimizers used for LoRA training are initialized at $0$, the optimization is biased towards smaller-magnitude solutions near the starting point, which are the global minima. In Section 4, we experimentally test this theory by fine-tuning LoRA with a non-zero initialization; indeed, we find there is an instance in this scenario, where the fine-tuning gets trapped in spurious local minima.

**Weight decay implicitly biases the optimization towards low-rank matrices.** Practical LoRA training typically employs weight decay (Hu et al., 2022; Dettmers et al., 2023), and it is shown in prior theoretical work that weight decay induces an implicit bias toward low-rank matrices. This makes it more likely for the LoRA training to converge to the low-rank global minimizer, rather than to a spurious local minima with high rank being $\sigma_r(X) > \frac{2\alpha}{\beta}\sigma_{r_\star}(X)$.

For deep *linear* networks, this implicit bias is characterized somewhat precisely.

**Theorem** (Informal, Theorem 3.2 of Wang & Jacot (2024))**.** *When training a deep linear network with positive weight decay, a sufficiently small learning rate, and a ground-truth teacher model with low effective rank, there is a positive probability of jumping from a high-rank critical point to a lower-rank one, but the probability of jumping back is zero.*

While the theory of Wang & Jacot (2024) does not immediately apply to general deep (non-linear) neural networks, it does provide meaningful insight into the implicit bias towards low rank. In the more general setup, Galanti et al. (2024) argues for a similar implicit bias. Adapting their arguments to LoRA training, we get the following statement.

**Lemma 1.** *Consider LoRA training with SGD with batch size $b$, learning rate $\mu$, and weight decay $\lambda > 0$. For any low-rank update $X = AB^\intercal$ of a weight matrix in the network, if the sequence of $X$-values throughout training converges to a matrix $\tilde{X}$, then $\tilde{X}$ is approximately low rank in the sense that for any $\varepsilon > 0$, there exists some $W$ with*

$$\left\|\frac{\tilde{X}}{\|\tilde{X}\|} - W\right\| < \varepsilon, \quad \text{rank}(W) \leq \frac{b\log(\varepsilon/4)}{\log(1 - \mu\lambda)}$$

We provide the proof of Lemma 1 in Appendix A.4.

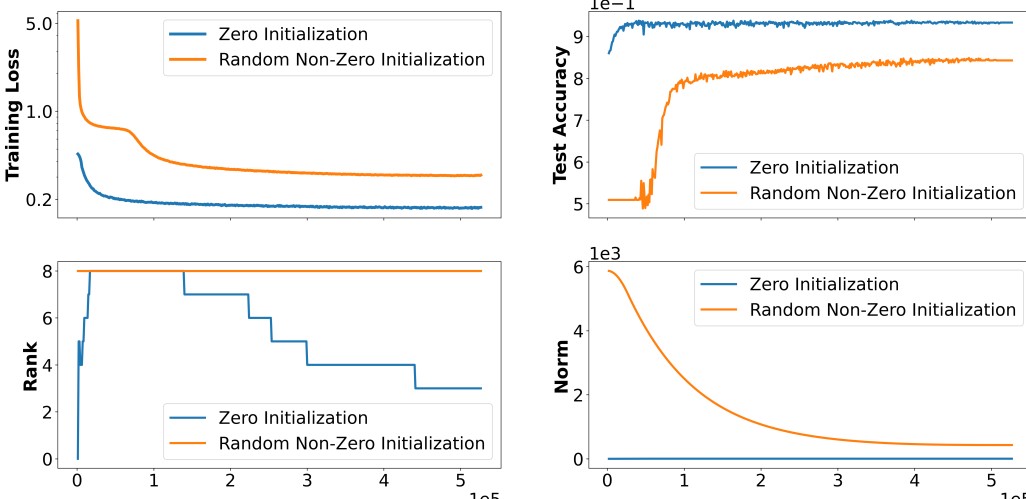

Figure 2: LoRA training converging to global minima with zero-initialization vs. spurious local minima with random non-zero initialization. **(y-axes)** Training loss, test accuracy, minimizer rank, and minimizer norm. **(x-axes)** Training steps.

## 4. Experiments

In this section, we validate our theory through real-world experiments. First, we verify our assumptions outlined in Section 2. Then, we present both the success and failure modes of LoRA fine-tuning, where training either converges to a low-rank, small-magnitude global minimizer or stuck on a high-rank, large-magnitude local minimizer.

**Experimental setup.** We conduct experiments on two tasks in NLP and vision. For the NLP task, we fine-tune a RoBERTA-base model (Zhuang et al., 2021) on a sentiment analysis task, using the SST-2 dataset (Socher et al., 2013) from the GLUE benchmark (Wang et al., 2018). For the vision task, we fine-tune a vision transformer (Dosovitskiy et al., 2021) on the CIFAR100 dataset (Krizhevsky, 2009). Both models have 12 attention layers, and we tune the query and value weights of each layer, following the prescription of Hu et al. (2022). We describe further details in Appendix C.

**Results: Verifying low-rank global minima exist.** First, we validate our assumption of a low-rank global minimum. We perform full fine-tuning on the nuclear norm regularized loss objective $\widehat{\mathcal{L}}_\lambda^{full}$ with varying values of weight decay $\lambda$. The results of Table 1 exhibit a clear decreasing trend on the rank of the global minimum as a function of $\lambda$. Notably, when $\lambda$ is set to values at least $0.001$, the resulting rank is lower than typical LoRA ranks (4, 8, or 16).

**Results: Verifying RSC and RSM.** Next, we verify our assumption of restricted strong convexity and smoothness. As it is infeasible to exactly compute $\alpha$ and $\beta$ values, we estimate them by Monte-Carlo sampling with 1000 samples within rank bound $r = 8, 16, 32, 64$, distance bound $D = 5$,

Table 2: RSC and RSM values for different ranks.

| Rank | 8 | 16 | 32 | 64 |
|---|---|---|---|---|
| $\beta/\alpha$ | 8.0249 | 18.7032 | 320.82 | N/A |
| $\alpha$ | 0.0061 | 0.0029 | 0.0002 | $-0.0445$ |
| $\beta$ | 0.0492 | 0.0539 | 0.0726 | 0.3371 |

and $\lambda = 0.01$. Table 2 presents the $\alpha$ and $\beta$ values for the largest $\beta/\alpha$ value across weight matrices. We see as $r$ increases, $\alpha$ decreases and $\beta$ increases. In fact, when $r$ is as large as 64, the requirement $\alpha > 0$ breaks, and our theory no longer applies. These results demonstrate that our assumption of $\alpha > 0$ and $\beta < \infty$ is plausible using a low LoRA rank $r$. This also suggests that reduced memory footprint is not the only benefit of using small $r$; the $\alpha, \beta$-values that determine the loss landscape also become more favorable with small $r$.

**Results: Validating main theorem.** Finally, we verify our main result through an illustrative example for the SST2 task with $\lambda = 0.01$ and $r = 8$. To clarify, our results prove that spurious local minima may not exist, but when they do, they exhibit high rank and large norm, being readily distinguishable from the global minimum and thereby avoidable through zero initialization. We present in Figure 2 that such spurious local minimum found by large random initialization indeed fails loudly in the sense that it has high rank, large magnitude, and poor generalization performance. We further demonstrate in Appendix C that spurious local minima isn't found in any smaller initializations.

## 5. Conclusion

In this work, we theoretically analyze LoRA fine-tuning and obtain a new type of result: that a second-order station-

ary point is either a global minimizer with low rank and small magnitude or is a spurious solution with high rank and large magnitude. Unlike previous analyses based on linearization, our approach relies on a general condition of restricted strong convexity and smoothness, which are conditions the experiments of Section 4 confirm to be practical. We further argue that zero-initialization and weight decay in LoRA training induce an implicit bias toward this low-rank small-magnitude region, explaining why LoRA typically converges to global minima in practice.

While the primary focus of this work is on establishing the theoretical convergence of LoRA, our framework possesses broader practical relevance. The properties of spurious local minima that we characterize may be used to diagnose and monitor the fine-tuning process. Furthermore, as our framework relies solely on the low-rank decomposition structure of LoRA and a few minimal assumptions, our theory applies to many LoRA variants, including LoRA+ (Hayou et al., 2024), rsLoRA (Kalajdzievski, 2023), PiSSA (Meng et al., 2024), and MiLoRA (Wang et al., 2025) as well.

Our results open several avenues for future work. One is to perform a more rigorous analysis of the implicit bias induced by weight decay and zero initialization. Another intriguing insight is that the restricted strong convexity and smoothness constants $\alpha$ and $\beta$ improve as the LoRA rank decreases, suggesting that smaller-rank parameterizations enjoy more favorable optimization landscapes. This observation contrasts with the modern wisdom of deep learning theory that overparameterization helps training and aligns with recent results indicating that overparameterization can slow down training (Xu & Du, 2023; Xiong et al., 2024). Exploring this phenomenon further is another promising direction.

## Acknowledgments

EKR was supported by the Samsung Science and Technology Foundation (Project Number SSTF-BA2101-02) and the National Research Foundation of Korea (NRF) grant funded by the Korean government (No.RS-2024-00421203). JK acknowledges the support from the Ilju foundation scholarship. We also thank Jaesung Park, Uijeong Jang, and Sunwoo Kim for providing valuable feedback.

## Impact statement

This paper advances the understanding of Low-rank Adaptation, contributing to the broader field of Machine Learning. There are many potential societal consequences of our work, none of which we feel must be specifically highlighted here.

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

# A. Omitted theorems and proofs

## A.1. Full proof for Theorem 1

Here we present the complete proof of Theorem 1. For simplicity, we write $\widehat{\mathcal{L}}(X)$ as $f(X)$, $\widehat{\mathcal{L}}^{\text{lora}}(A, B)$ as $g(A, B)$.

Set $(A, B) \in \mathbb{R}^{m \times r} \times \mathbb{R}^{n \times r}$ to be a SOSP of $f_\lambda$, $AB^{\intercal}$ as $X$, the compact SVD of $X$ as $L_X \Sigma_X R_X^{\intercal}$, and $\sigma_i$ as the $i$-th (largest) singular value of $X$. From the first and second-order optimality of $\mathcal{L}_\lambda(A, B)$, we acquire the following properties:

1. $0 = \nabla_A g(A, B) = \nabla f(AB^{\intercal}) \cdot B + \lambda A$

2. $0 = \nabla_B g(A, B) = \nabla f(AB^{\intercal})^{\intercal} \cdot A + \lambda B$

3. $\nabla^2 g(A, B)[(U, V), (U, V)] = 2\langle \nabla f(X), UV^{\intercal} \rangle + \nabla^2 f(X)[AV^{\intercal} + UB^{\intercal}, AV^{\intercal} + UB^{\intercal}] + \lambda(\|U\|_F^2 + \|V\|_F^2) \geq 0$, $\forall (U, V) \in \mathbb{R}^{m \times r} \times \mathbb{R}^{n \times r}$

By equations 1 and 2,
$$-A^{\intercal} \nabla f(AB^{\intercal})B = \lambda A^{\intercal} A = \lambda B^{\intercal} B,$$

so if $\lambda > 0$ then $A^{\intercal} A = B^{\intercal} B$. Therefore by Lemma B.1, we can set $A = L_X \Sigma_X^{1/2} W$, $B = R_X \Sigma_X^{1/2} W$ for some orthogonal matrix $W$. Plugging $A, B$ back into properties 1 and 2 gives us
$$\nabla f(X) \cdot R_X = -\lambda L_X, \quad \nabla f(X)^{\intercal} \cdot L_X = -\lambda R_X.$$

If $\lambda = 0$, then apparently $\nabla f(X) \cdot R_X = 0$ and $\nabla f(X)^{\intercal} \cdot L_X = 0$, so the equation above holds regardless of $\lambda$. Now by Lemma B.2, $\nabla f(X)$ can be represented as
$$\nabla f(X) = -\lambda L_X R_X^{\intercal} + S, \quad S = \tilde{L}_X \tilde{\Sigma}_X \tilde{R}_X^{\intercal},$$

for some diagonal matrix $\tilde{\Sigma}_X$ and some $\tilde{L}_X, \tilde{R}_X$ that $\begin{bmatrix} L_X & \tilde{L}_X \end{bmatrix}$ and $\begin{bmatrix} R_X & \tilde{R}_X \end{bmatrix}$ are orthogonal. Now we will show that the spectral norm of $S$ is bounded by $\lambda + \beta \sigma_r$.

We first consider the case where $\text{rank}(X) = r$, or $\sigma_r(X) > 0$. Setting $u_\star, v_\star$ as the top singular vectors of $S$, and plugging in $(U, V) = (-u_\star u_r^{\intercal} A, v_\star v_r^{\intercal} B)$ into property 3, we achieve the following inequality.

$$\begin{aligned}
\nabla^2 f(X)[Xv_r v_\star^{\intercal} - u_\star u_r^{\intercal} X, Xv_r v_\star^{\intercal} - u_\star u_r^{\intercal} X] + \lambda \left( \|U\|_F^2 + \|V\|_F^2 \right) &\geq 2\langle \nabla f(X), u_\star u_r^{\intercal} X v_r v_\star^{\intercal} \rangle \\
&= 2\langle -\lambda L_X R_X^{\intercal} + S, u_\star u_r^{\intercal} X v_r v_\star^{\intercal} \rangle \\
&= 2\sigma_r \langle -\lambda L_X R_X^{\intercal} + S, u_\star v_\star^{\intercal} \rangle \\
&= 2\sigma_r \langle S, u_\star v_\star^{\intercal} \rangle = 2\sigma_r \|S\|_2.
\end{aligned}$$

Where
$$\begin{aligned}
\|U\|_F^2 + \|V\|_F^2 &= tr(u_\star u_r^{\intercal} A A^{\intercal} u_r u_\star^{\intercal}) + tr(v_\star v_r^{\intercal} B B^{\intercal} v_r v_\star^{\intercal}) \\
&= tr(u_\star u_r^{\intercal} L_X \Sigma_X L_X^{\intercal} u_r u_\star^{\intercal}) + tr(v_\star v_r^{\intercal} R_X \Sigma_X R_X^{\intercal} v_r v_\star^{\intercal}) \\
&= \sigma_r \cdot (tr(u_\star u_\star^{\intercal}) + tr(v_\star v_\star^{\intercal})) = 2\sigma_r
\end{aligned}$$

Now the restricted smoothness of $f$ yields the following inequality.
$$\begin{aligned}
\nabla^2 f(X)[Xv_r v_\star^{\intercal} - u_\star u_r^{\intercal} X, Xv_r v_\star^{\intercal} - u_\star u_r^{\intercal} X] &\leq \beta \|Xv_r v_\star^{\intercal} - u_\star u_r^{\intercal} X\|_F^2 \\
&= \beta \sigma_r^2 \|u_r v_\star^{\intercal} - u_\star v_r^{\intercal}\|_F^2 \leq 2\beta \sigma_r^2
\end{aligned}$$

Combining the two inequalities results in
$$\|S\|_2 \leq \beta \sigma_r + \lambda$$

Now we see the case where $\text{rank}(X) = r' < r$, or $\sigma_r(X) = 0$. Since $\text{rank}(X) < r$, $A$ and $B$ are also rank deficient, so we can find a unit vector $w$ that $Aw = 0$. Now
$$Aw = 0 \Rightarrow A^{\intercal} A w = 0 \Rightarrow B^{\intercal} B w = 0 \Rightarrow w B^{\intercal} B w = 0 \Rightarrow \|Bw\| = 0$$

so we have $Bw = 0$ as well. Now for any unit vector $u \in \mathbb{R}^m$, $v \in \mathbb{R}^n$, plugging in $(U, V) = (uw^{\mathsf{T}}, -vw^{\mathsf{T}})$ into property 3 results in

$$\langle \nabla f(X), uv^{\mathsf{T}} \rangle \leq \lambda$$

Since this holds for any unit vector $u, v$, we have $\|\nabla f(X)\|_2 \leq \lambda$. Since $\nabla f(X) = -\lambda L_X R_X^{\mathsf{T}} + S$, this implies $\|S\|_2 \leq \lambda$ as well.

Now that we know $\|S\|_2 \leq \beta \sigma_r + \lambda$ for all cases, we will find what this implies. Consider the subgradient $\partial(\lambda \|X\|_*)$, which we know the explicit form as

$$\partial(\lambda \|X\|_*) = \left\{ G \in \mathbb{R}^{m \times n} \;\middle|\; G = \lambda L_X R_X^{\mathsf{T}} + W,\; L_X^{\mathsf{T}} W = 0,\; W R_X = 0,\; \|W\|_2 \leq \lambda \right\}$$

Since $S$ also satisfies $L_X^{\mathsf{T}} S = 0$ and $S R_X = 0$, there exists an element $g \in \partial(\lambda \|X\|_*)$ that $\|g + \nabla f(X)\|_2 \leq \beta \sigma_r$. Now let $Z = (g + \nabla f(X))$. For any positive real number $\kappa > 0$ that $\kappa \times \beta \sigma_r \leq \sigma_{r_\star}$, the singular vectors of $\kappa Z$ are orthogonal to those of $X$ and $\|\kappa Z\|_2 \leq \sigma_{r_\star}$, so the top $r_\star$ singular vectors of $X - \kappa Z$ coincide with those of $X$. Therefore, by the Eckart–Young–Mirsky theorem we see

$$X^{r_\star} \in \underset{\text{rank}(Y) \leq r_\star}{argmin} \|Y - (X - \kappa Z)\|_F^2$$

where $X^{r_\star}$ is a projection of $X$ onto the set of matrices of rank $r_\star$ or less. Since $\text{rank}(X_\star) = r_\star$, we can plug in $X_\star$ into $Y$ here, resulting in

$$\|X^{r_\star} - X + \kappa Z\|_F^2 \leq \|X_\star - X + \kappa Z\|_F^2$$

which is again equivalent to

$$\|X^{r_\star} - X\|_F^2 + 2k\langle X^{r_\star} - X, Z \rangle \leq \|X_\star - X\|_F^2 + 2k\langle X_\star - X, Z \rangle \tag{3}$$

Here we actually know that $\langle X^{r_\star} - X, Z \rangle = 0$, since the singular vectors of $X$ and $Z$ are orthogonal. Now by restricted convexity at $X_\star$, we have

$$\langle X - X_\star, \nabla f(X) - \nabla f(X_\star) \rangle \geq \alpha \|X - X_\star\|_F^2 \tag{4}$$

Since $X_\star$ is the global minimizer of $f_\lambda(X) = f(X) + \lambda \|X\|_\star$ and the nuclear norm is lower semi-continuous, from Theorem 3.1 of (Mordukhovich & Shao, 1995) we have $-\nabla f(X_\star) \in \partial(\lambda \|X_\star\|_*)$. We know by definition that $g \in \partial(\lambda \|X\|_*)$, so the property of the subgradient implies

$$\langle X - X_\star, g - (-\nabla f(X_\star)) \rangle \geq 0 \tag{5}$$

Summing up the inequalities (3),(4),(5) we have

$$(2\kappa\alpha - 1)\|X_\star - X\|_F^2 + \|X^{r_\star} - X\|_F^2 \leq 0$$

Since this inequality holds for any positive real number $\kappa$ that $\kappa \times \beta \sigma_r \leq \sigma_{r_\star}$, if $\sigma_r = 0$ we can select an arbitrarily large $\kappa$, resulting in $X_\star = X$. If $\sigma_r \neq 0$, we can plug in $\kappa = \frac{\sigma_{r_\star}}{\sigma_r}$, resulting in $X = X_\star$ if $1 \geq 2\kappa\alpha$ and

$$\|X_\star - X\|_F^2 \geq \frac{\|X^{r_\star} - X\|_F^2}{1 - 2\kappa\alpha} = \frac{\sum_{s=r_\star+1}^{r} \sigma_s^2}{1 - 2\kappa\alpha}$$

otherwise. Therefore our statement is proven.

### A.2. Proof of Theorem 3

Here we present the proof of Theorem 3.

*Proof.* We can proceed with the identical reasoning and notations with the proof of Theorem 1 in A.1, up to defining $Z = g + \nabla f(X)$ with $\|Z\|_2 \leq \beta \sigma_r$. Now denoting the true global minimizer as $X_\star$, we have a rank $r_\star$ approximate global minimizer $X_\star^{r_\star}$ with $\|X_\star - X_\star^{(\delta)}\|_F \leq \delta$. We first prove that $\sigma_r(X) \geq \varepsilon \cdot \frac{\alpha}{2\beta\sqrt{r}}$ if $X$ is an $\varepsilon$-spurious local minima, which requires independent reasoning, and then we prove the remaining results analogously to Theorem 1.

**Step 1.** Fix some constant $c > 0$, and define $r' = \min\{\gamma \,|\, \sigma_\gamma(X) < c, \, 1 \leq \gamma \leq r\}$. Now setting $\kappa$ as a positive real number that $\kappa \times \beta\sigma_r < c$. By the Eckart–Young–Mirsky theorem we see

$$\underset{\mathrm{rank}(Y)\leq r_\star}{argmin} \|Y - (X - \kappa Z)\|_F^2 = (X - \kappa Z)^{r_\star}$$

where $(X - \kappa Z)^{r_\star}$ is the projection of $X - \kappa Z$ onto the set of matrices of rank $r_\star$ or less. Since $\|\kappa Z\|_2 < c$, by definition of $r'$ the 1-th to $r$-th singular vectors of $X - \kappa Z$ coincide with those of $X$, while the subsequent singular values are all smaller than $c$. This implies

$$\|X^{r'} - (X - \kappa Z)^{r_\star}\|_F^2 \leq c^2 \cdot \max\{r_\star - r', 0\}$$

If $r_\star < r'$, we are done, so we can consider only the case where $r_\star - r' \geq 0$. Combining the two relations, we have

$$\|X^{r'} - (X - \kappa Z)\|_F \leq \|X^{r'} - (X - \kappa Z)^{r_\star}\|_F + \|(X - \kappa Z)^{r_\star} - (X - \kappa Z)\|_F$$
$$\leq c\sqrt{r_\star - r'} + \|X_\star^{(\delta)} - (X - \kappa Z)\|_F$$

and squaring each sides, this expands to

$$\|X^{r'} - X\|_F^2 \leq c^2(r_\star - r') + 2c\sqrt{r_\star - r'}\|X^{r'} - (X - \kappa Z)\|_F + \|X_\star^{(\delta)} - X\|_F^2 + 2k\langle X_\star^{(\delta)} - X^{r'}, Z\rangle$$

Now in the same way with A.1, due to the restricted convexity and the subgradient property we have

$$2\alpha\kappa\|X - X_\star\|_F^2 \leq 2k\langle X - X_\star, Z\rangle.$$

Adding the two up, we have

$$(2\alpha\kappa - 1)\|X - X_\star\|_F^2 + \|X^{r'} - X\|_F^2 \leq c^2(r_\star - r) + 2c\sqrt{r_\star - r'}\|X^{r'} - (X - \kappa Z)\|_F + (\|X_\star^{r_\star} - X\|_F^2 - \|X_\star - X\|_F^2)$$
$$+ 2k\langle X_\star^{r_\star} - X_\star, Z\rangle + 2k\langle X - X^{r'}, Z\rangle$$

which again simplifies to

$$(2\alpha\kappa - 1)\|X - X_\star\|_F^2 - 2\delta\|X_\star - X\|_F \leq 3c^2(r - r') + 2c\delta\sqrt{r - r_\star},$$

or equivalently

$$\|X - X_\star\|_F \leq \frac{\delta}{2\alpha\kappa - 1} + \sqrt{\left(\frac{\delta}{2\alpha\kappa - 1}\right)^2 + 3c^2(r - r') + 2c\delta\sqrt{r - r_\star}}.$$

Therefore, setting $c = \frac{\varepsilon}{2\sqrt{r - r'}}$ and $\kappa = \frac{c}{\beta\sigma_r}$ (or any large number if $\sigma_r = 0$), if $\sigma_r(X) < \frac{\alpha c}{\beta}$, then $\delta = o(\varepsilon^3)$ implies $\|X - X_\star\|_F < \varepsilon$, making $X$ an $\varepsilon$-global minimizer. Thus if $X$ is a spurious local minima, then $\sigma_r(X) \geq \frac{\alpha}{2\beta\sqrt{r}} \cdot \varepsilon$

**Step 2.** Similarly to Theorem 1, define $\kappa$ as a positive real number that $\kappa \cdot \beta\sigma_r < \sigma_{r_\star}$, implying $\|\kappa Z\| \leq \sigma_{r_\star}$. Then

$$X^{r_\star} \in \underset{\mathrm{rank}(Y)\leq r_\star}{argmin} \|Y - (X - \kappa Z)\|_F^2$$

and $\mathrm{rank}(X_\star^{(\delta)}) = r_\star$ so

$$\|X^{r_\star} - X + \kappa Z\|_F^2 \leq \|X_\star^{(\delta)} - X + \kappa Z\|_F^2$$

which is equivalent to

$$\|X^{r_\star} - X\|_F^2 \leq \|X_\star^{r_\star} - X\|_F^2 + 2k\langle X_\star^{r_\star} - X, Z\rangle.$$

Now the relations (4),(5) from the proof of Theorem 1 still apply here so adding these we have

$$\alpha\|X - X_\star^{r_\star}\|_F^2 \leq \kappa\langle X - X_\star, Z\rangle$$

so adding the two inequalities, we obtain

$$2\kappa\alpha\|X - X_\star^{r_\star}\|_F^2 - \|X - X_\star^{r_\star}\|_F^2 + \|X^{r_\star} - X\|_F^2 \leq 2k\langle X_\star^{r_\star} - X_\star, Z\rangle.$$

Here we can upper bound $\langle X_\star^{r_\star} - X_\star, Z \rangle$ by $2\sigma_{r_\star} \cdot \|X_\star^{(\delta)} - X_\star\|_*$, by the duality of the nuclear norm and spectral norm and $\|\kappa Z\|_2 \leq \sigma_{r_\star}$. $\sigma_{r_\star}$ is again bounded by $\frac{D}{\sqrt{r_\star}}$ since we are looking at $\|X - X_\star\|_F \leq \|X\|_F + \|X_\star\|_F \leq 2D$, and $\|X_\star^{(\delta)} - X_\star\|_* \leq \sqrt{r - r_\star} \cdot \|X_\star^{(\delta)} - X_\star\|_F \leq \delta \times \sqrt{r - r_\star}$ by the Cauchy-Schwartz inequality. Therefore we have

$$2\kappa\alpha\|X - X_\star^{r_\star}\|_F^2 - \|X - X_\star^{r_\star}\|_F^2 + \|X^{r_\star} - X\|_F^2 \leq 2\delta D\sqrt{\frac{r - r_\star}{r_\star}}.$$

We can expand this inequality, using $X - X_\star^{r_\star} = (X - X_\star) + (X_\star - X_\star^{r_\star})$ as

$$(2\kappa\alpha - 1)\|X_\star - X\|_F^2 + \|X^{r_\star} - X\|_F^2 \leq 2\langle X - X_\star, X_\star - X_\star^{r_\star}\rangle + \delta D\sqrt{\frac{r - r_\star}{r_\star}} \leq 2\delta\|X - X_\star\|_F + \delta 2D\sqrt{\frac{r - r_\star}{r_\star}}.$$

Solving the quadratic inequality about $\|X_\star^{r_\star} - X\|_F$, if $2\kappa\alpha - 1 > \varepsilon$ the discriminant is

$$\delta^2 + 4(2\kappa\alpha - 1)\left(2\delta D\sqrt{\frac{r - r_\star}{r_\star}} - \|X^{r_\star} - X\|_F^2\right).$$

Now we know that if $X$ is an $\varepsilon$-spurious local minima then $\|X^{r_\star} - X\|_F \geq \sigma_r \geq \frac{\alpha}{2\beta\sqrt{r}} \cdot \varepsilon$, so the discriminant is negative since $\delta = o(\varepsilon^2)$ and therefore there would be no spurious local minima. If $2\kappa\alpha - 1 < 0$, we would see

$$\|X_\star - X\|_F \leq \sqrt{\frac{\|X^{r_\star} - X\|_F^2 - (2\delta D\sqrt{\frac{r - r_\star}{r_\star}} - \frac{\delta^2}{1 - 2\kappa\alpha})}{1 - 2\kappa\alpha}} - \frac{\delta}{1 - 2\kappa\alpha}$$

resulting in

$$\|X_\star - X\|_F \leq \sqrt{\frac{\|X^{r_\star} - X\|_F^2 - \varepsilon^3}{1 - 2\kappa\alpha}} - \varepsilon^2$$

given $\delta = o(\varepsilon^3)$ $\qquad\qquad\qquad\qquad\qquad\qquad\qquad\qquad\qquad\qquad\qquad\qquad\qquad\qquad\qquad\qquad\quad\square$

### A.3. Theorems for the multiple matrix case

In Section 3, we presented most theorems as a result for tuning a single matrix for clarity. Here we explicitly portray the natural extension to the multi matrix case for the theorems and method of proof. First, we provide the proof of Theorem 2, the multi matrix case without error.

*Proof.* As in the proofs above, we write $\widehat{\mathcal{L}}(\mathbf{X})$ as $f(\mathbf{X})$, $\widehat{\mathcal{L}}^{\text{lora}}(\mathbf{A}, \mathbf{B})$ as $g(\mathbf{A}, \mathbf{B})$ for simplicity. Set $(\mathbf{A}, \mathbf{B})$ to be a SOSP of $f_\lambda$, and $\mathbf{A}\mathbf{B}^\intercal$ as $\mathbf{X}_\square = (X_\square^{(1)}, \ldots, X_\square^{(L)})$. From the first and second-order optimality of $\mathcal{L}_\lambda(\mathbf{A}, \mathbf{B})$, we acquire the following properties:

1. $0 = \nabla_{\mathbf{A}} g(\mathbf{A}, \mathbf{B}) = \nabla f(\mathbf{A}\mathbf{B}^\intercal) \cdot \mathbf{B} + \lambda\mathbf{A}$

2. $0 = \nabla_{\mathbf{B}} g(\mathbf{A}, \mathbf{B}) = \nabla f(\mathbf{A}\mathbf{B}^\intercal)^\intercal \cdot \mathbf{A} + \lambda\mathbf{B}$

3. $\nabla^2 g(\mathbf{A}, \mathbf{B})[(\mathbf{U}, \mathbf{V}), (\mathbf{U}, \mathbf{V})] = 2\langle \nabla f(\mathbf{X}), \mathbf{U}\mathbf{V}^\intercal\rangle + \nabla^2 f(\mathbf{X})[\mathbf{A}\mathbf{V}^\intercal + \mathbf{U}\mathbf{B}^\intercal, \mathbf{A}\mathbf{V}^\intercal + \mathbf{U}\mathbf{B}^\intercal] + \lambda(\|\mathbf{U}\|_F^2 + \|\mathbf{V}\|_F^2) \geq 0$

Recall that for a tuple of matrices

$$\mathbf{C} = (C^{(1)}, \ldots, C^{(L)}), \quad , (D^{(1)}, \ldots, D^{(L)})$$

we write the product of the tuples as

$$\mathbf{C}\mathbf{D}^\intercal = (C^{(1)}(D^{(1)})^\intercal, \ldots, C^{(L)}(D^{(L)})^\intercal),$$

the scalar product as

$$k\mathbf{C} = (kC^{(1)}, \ldots, kC^{(L)}),$$

and the Frobenius norm as

$$\|\mathbf{C}\|_F = \sqrt{\sum_{l=1}^{L} \|C^{(l)}\|_F^2}.$$

We interpret the gradient $\nabla f(\mathbf{AB}^\intercal)$ as a tuple with the same shape with $\mathbf{AB}^\intercal$, and the Hessian $\nabla^2 f(\mathbf{X})$ as a $(m_1 n_1 + \cdots + m_L n_L) \times (m_1 n_1 + \cdots + m_L n_L)$ block matrix consisted of $m_i n_i \times m_j n_j$ blocks $\nabla_{i,j}^2 f(X)$.

By looking at only the $l$th matrices from the equations 1,2, we have

1. $0 = \nabla_{A^{(l)}} g(A^{(l)}, B^{(l)}) = \nabla_l f(\mathbf{AB}^\intercal) \cdot B^{(l)} + \lambda A^{(l)}$

2. $0 = \nabla_{B^{(l)}} g(A^{(l)}, B^{(l)}) = \nabla_l f(\mathbf{AB}^\intercal)^\intercal \cdot A^{(l)} + \lambda B^{(l)}$

Furthermore, inputting $\mathbf{U}, \mathbf{V}$ with

$$\mathbf{U} = \left(0, \ldots, U^{(l)}, \ldots, 0\right), \quad , \mathbf{V} = \left(0, \ldots, V^{(l)}, \ldots, 0\right)$$

for some $U^{(l)}, V^{(l)}$ we see

3. $2\langle \nabla_l f(\mathbf{X}), U^{(l)} V^{(l)\intercal} \rangle + \nabla_{l,l}^2 f(\mathbf{X})[A^{(l)}(V^{(l)})^\intercal + U^{(l)}(B^{(l)})^\intercal, A^{(l)}(V^{(l)})^\intercal + U^{(l)}(B^{(l)})^\intercal] + \lambda(\|U^{(l)}\|_F^2 + \|V^{(l)}\|_F^2) \geq 0$

Now we can proceed similarly with the single matrix case. By the new versions of equations 1 and 2,

$$-(A^{(l)})^\intercal \nabla f(\mathbf{AB}^\intercal) B^{(l)} = \lambda (A^{(l)})^\intercal A^{(l)} = \lambda (B^{(l)})^\intercal B^{(l)},$$

so if $\lambda > 0$ then $(A^{(l)})^\intercal A^{(l)} = (B^{(l)})^\intercal B^{(l)}$. Now denote the SVD of $X^{(l)}$ as $L_X^{(l)} \Sigma_X^{(l)} (R_X^{(l)})^\intercal$, and $\sigma_i^{(l)}$ as the $i$-th (largest) singular value of $X^{(l)}$. By Lemma B.1, we can set $A^{(l)} = L_X^{(l)} (\Sigma_X^{(l)})^{1/2} W$, $B^{(l)} = R_X^{(l)} (\Sigma_X^{(l)})^{1/2} W$ for some orthogonal matrix $W$. Plugging $A^{(l)}, B^{(l)}$ back into properties 1 and 2 gives us

$$\nabla_l f(\mathbf{X}) \cdot R_X^{(l)} = -\lambda L_X^{(l)}, \quad \nabla f(\mathbf{X})^\intercal \cdot L_X^{(l)} = -\lambda R_X^{(l)}.$$

If $\lambda = 0$, then apparently $\nabla_l f(\mathbf{X}) \cdot R_X^{(l)} = 0$ and $\nabla_l f(\mathbf{X})^\intercal \cdot L_X^{(l)} = 0$, so the equation above holds regardless of $\lambda$. Now by Lemma B.2, $\nabla_l f(X)$ can be represented as

$$\nabla_l f(\mathbf{X}) = -\lambda L_X^{(l)} R_X^{(l)\intercal} + S, \quad S = \tilde{L}_X^{(l)} \tilde{\Sigma}_X (\tilde{R_X}^{(l)})^\intercal,$$

for some diagonal matrix $\tilde{\Sigma}_X$ and some $\tilde{L}_X^{(l)}, \tilde{R}_X^{(l)}$ that $\left[L_X^{(l)} \ \tilde{L}_X^{(l)}\right]$ and $\left[R_X^{(l)} \ \tilde{R}_X^{(l)}\right]$ are orthogonal. Now we will show that the spectral norm of $S$ is bounded by $\lambda + \beta^{(l)} \sigma_r^{(l)}$.

We first consider the case where $\text{rank}(X^{(l)}) = r$, or $\sigma_r(X^{(l)}) > 0$. Setting $u_\star, v_\star$ as the top singular vectors of $S$, and plugging in $(U^{(l)}, V^{(l)}) = (-u_\star u_r^\intercal A^{(l)}, v_\star v_r^\intercal B^{(l)})$ into property 3, we achieve the following inequality.

$$\nabla_{l,l}^2 f(\mathbf{X})[X^{(l)} v_r v_\star^\intercal - u_\star u_r^\intercal X^{(l)}, X^{(l)} v_r v_\star^\intercal - u_\star u_r^\intercal X^{(l)}] + \lambda \left(\|U^{(l)}\|_F^2 + \|V^{(l)}\|_F^2\right) \geq 2\langle \nabla_l f(\mathbf{X}^{(1)}), u_\star u_r^\intercal X^{(l)} v_r v_\star^\intercal \rangle$$

$$= 2\langle -\lambda L_X^{(l)} R_X^{(l)\intercal} + S, u_\star u_r^\intercal X^{(l)} v_r v_\star^\intercal \rangle$$

$$= 2\sigma_r^{(l)} \langle -\lambda L_X^{(l)} R_X^{(l)\intercal} + S, u_\star v_\star^\intercal \rangle$$

$$= 2\sigma_r^{(l)} \langle S, u_\star v_\star^\intercal \rangle = 2\sigma_r^{(l)} \|S\|_2.$$

Where

$$\|U^{(l)}\|_F^2 + \|V^{(l)}\|_F^2 = tr(u_\star u_r^\intercal A^{(l)} (A^{(l)})^\intercal u_r u_\star^\intercal) + tr(v_\star v_r^\intercal B^{(l)} (B^{(l)})^\intercal v_r v_\star^\intercal)$$

$$= tr(u_\star u_r^\intercal L_X^{(l)} \Sigma_X^{(l)} L_X^{(l)\intercal} u_r u_\star^\intercal) + tr(v_\star v_r^\intercal R_X^{(l)} \Sigma_X^{(l)} R_X^{(l)\intercal} v_r v_\star^\intercal)$$

$$= \sigma_r^{(l)} \cdot (tr(u_\star u_\star^\intercal) + tr(v_\star v_\star^\intercal)) = 2\sigma_r^{(l)}$$

Now the restricted smoothness of $f$ yields the following inequality.

$$\nabla^2 f(X^{(l)})[X^{(l)}v_r v_\star^\mathsf{T} - u_\star u_r^\mathsf{T} X^{(l)}, X^{(l)}v_r v_\star^\mathsf{T} - u_\star u_r^\mathsf{T} X^{(l)}] \leq \beta^{(l)} \left\| X^{(l)}v_r v_\star^\mathsf{T} - u_\star u_r^\mathsf{T} X^{(l)} \right\|_F^2$$

$$= \beta^{(l)} \sigma_r^{(l)2} \|u_r v_\star^\mathsf{T} - u_\star v_r^\mathsf{T}\|_F^2 \leq 2\beta^{(l)} \sigma_r^{(l)2}$$

Combining the two inequalities results in

$$\|S\|_2 \leq \beta^{(l)} \sigma_r^{(l)} + \lambda.$$

Now we see the case where $\mathrm{rank}(X^{(l)}) = r' < r$, or $\sigma_r(X^{(l)}) = 0$. Since $\mathrm{rank}(X^{(l)}) < r$, $A^{(l)}$ and $B^{(l)}$ are also rank deficient, so we can find a unit vector $w$ that $A^{(l)}w = 0$. Now

$$A^{(l)}w = 0 \Rightarrow (A^{(l)})^\mathsf{T} A^{(l)}w = 0 \Rightarrow (B^{(l)})^\mathsf{T} B^{(l)}w = 0 \Rightarrow w(B^{(l)})^\mathsf{T} B^{(l)}w = 0 \Rightarrow \left\| B^{(l)}w \right\| = 0$$

so we have $B^{(l)}w = 0$ as well. Now for any unit vector $u \in \mathbb{R}^m, v \in \mathbb{R}^n$, plugging in $(U, V) = (uw^\mathsf{T}, -vw^\mathsf{T})$ into property 3 results in

$$\langle \nabla_l f(\mathbf{X}), uv^\mathsf{T} \rangle \leq \lambda$$

Since this holds for any unit vector $u, v$, we have $\|\nabla_l f(\mathbf{X})\|_2 \leq \lambda$. Since $\nabla_l f(\mathbf{X}) = -\lambda L_X^{(l)}(R_X^{(l)})^\mathsf{T} + S$, this implies $\|S\|_2 \leq \lambda$ as well.

Now that we know $\|S\|_2 \leq \beta^{(l)} \sigma_r^{(l)} + \lambda$ for all cases, we will find what this implies. Consider the subgradient $\partial(\lambda \|X^{(l)}\|_*)$, which we know the explicit form as

$$\partial(\lambda \|X^{(l)}\|_*) = \left\{ G \in \mathbb{R}^{m \times n} \,\middle|\, G = \lambda L_X^{(l)}(R_X^{(l)})^\mathsf{T} + W, \ (L_X^{(l)})^\mathsf{T} W = 0, \ W R_X^{(l)} = 0, \ \|W\|_2 \leq \lambda \right\}$$

Since $S$ also satisfies $(L_X^{(l)})^\mathsf{T} S = 0$ and $S R_X^{(l)} = 0$, there exists an element $g \in \partial(\lambda \|X^{(l)}\|_*)$ that $\|g + \nabla_l f(\mathbf{X})\|_2 \leq \beta^{(l)} \sigma_r^{(l)}$. Now let $Z = (g + \nabla_l f(\mathbf{X}))$. For any positive real number $\kappa > 0$ that $\kappa \times \beta^{(l)} \sigma_r^{(l)} \leq \sigma_{r_\star}^{(l)}$, the singular vectors of $\kappa Z$ are orthogonal to those of $X^{(l)}$ and $\|\kappa Z\|_2 \leq \sigma_{r_\star}^{(l)}$, so the top $r_\star$ singular vectors of $X^{(l)} - \kappa Z$ coincide with those of $X^{(l)}$. Therefore, by the Eckart–Young–Mirsky theorem we see

$$(X^{(l)})^{r_\star} \in \underset{\mathrm{rank}(Y) \leq r_\star}{\arg\min} \|Y - (X^{(l)} - \kappa Z)\|_F^2$$

where $(X^{(l)})^{r_\star}$ is a projection of $X^{(l)}$ onto the set of matrices of rank $r_\star$ or less. Since $\mathrm{rank}(X_\star^{(l)}) \leq r_\star$, we can plug in $X_\star^{(l)}$ into $Y$ here, resulting in

$$\|(X^{(l)})^{r_\star} - X^{(l)} + \kappa Z\|_F^2 \leq \|X_\star^{(l)} - X^{(l)} + \kappa Z\|_F^2$$

which is again equivalent to

$$\|(X^{(l)})^{r_\star} - X^{(l)}\|_F^2 + 2k\langle (X^{(l)})^{r_\star} - X^{(l)}, Z \rangle \leq \|X_\star^{(l)} - X^{(l)}\|_F^2 + 2k\langle X_\star^{(l)} - X^{(l)}, Z \rangle \tag{6}$$

Here we actually know that $\langle (X^{(l)})^{r_\star} - X^{(l)}, Z \rangle = 0$, since the singular vectors of $X^{(l)}$ and $Z$ are orthogonal. Now by restricted convexity at $X_\star^{(l)}$, we have

$$\langle X^{(l)} - X_\star^{(l)}, \nabla_l f(\mathbf{X}) - \nabla_l f(\mathbf{X}_\star) \rangle \geq \alpha^{(l)} \left\| X^{(l)} - X_\star^{(l)} \right\|_F^2 \tag{7}$$

Since $X_\star$ is the global minimizer of $f_\lambda(\mathbf{X}) = f(\mathbf{X}) + \lambda \|\mathbf{X}\|_\star$ and the nuclear norm is lower semi-continuous, from Mordukhovich & Shao (1995, Theorem 3.1) we have $-\nabla_l f(\mathbf{X}_\star) \in \partial(\lambda \|X_\star^{(l)}\|_*)$. We know by definition that $g \in \partial(\lambda \|X^{(l)}\|_*)$, so the property of the subgradient implies

$$\langle X^{(l)} - X_\star^{(l)}, g - (-\nabla_l f(X_\star)) \rangle \geq 0 \tag{8}$$

Summing up the inequalities (6),(7),(8) we have

$$(2\kappa \alpha^{(l)} - 1)\|X_\star^{(l)} - X^{(l)}\|_F^2 + \|(X^{(l)})^{r_\star} - X^{(l)}\|_F^2 \leq 0$$

Since this inequality holds for any positive real number $\kappa$ that $\kappa \times \beta^{(l)}\sigma_r^{(l)} \le \sigma_{r_\star}$, if $\sigma_r^{(l)} = 0$ we can select an arbitrarily large $\kappa$, resulting in $X_\star^{(l)} = X^{(l)}$. If $\sigma_r^{(l)} \ne 0$, we can plug in $\kappa = \frac{\sigma_{r_\star}^{(l)}}{\sigma_r^{(l)}}$, resulting in $X^{(l)} = X_\star^{(l)}$ if $1 \ge 2\kappa\alpha^{(l)}$ and

$$\|X_\star^{(l)} - X^{(l)}\|_F^2 \ge \frac{\|(X^{(l)})^{r_\star} - X^{(l)}\|_F^2}{1 - 2\kappa\alpha^{(l)}}$$

otherwise. Therefore, if $\sigma_r(X^{(l)}) \ge \frac{\alpha^{(l)}}{2\beta^{(l)}} \cdot \sigma_{r_\star}(X^{(l)})$ for all $1 \le l \le L$ then $X_\star^{(l)} = X^{(l)}$ for all $1 \le l \le L$, resulting in $\mathbf{X} = \mathbf{X}_\star$. If not, there must exist some $l$ satisfying the inequality above. Therefore our statement is proved. $\qquad \square$

Now we present a Master Theorem, applicable for the most general case with multiple matrices and an approximately low rank solution. We first analogously define a $\delta$-global minimizer for the multiple matrix case:

We say $\mathbf{X}_\star^{(\delta)}$ is a $\delta$-*global minimizer of full fine-tuning* if

$$\|(X_\star^{(l)})^{(\delta)} - X_\star^{(l)}\|_F \le \delta , \quad 1 \le l \le L$$

where $\mathbf{X}_\star$ is an 'exact' global minimizer of $\widehat{\mathcal{L}}^{\mathrm{full}}$. Based on this definition, we present the analogous result.

**Theorem A.1.** *(Master Theorem)* *Let $\varepsilon > 0$ and $\lambda \ge 0$. Assume the full fine-tuning loss $\widehat{\mathcal{L}}_\lambda^{\mathrm{full}}$ has an exact global minimizer $\mathbf{X}_\star$, and a rank-$r_\star$ $\delta$-global minimizer $\mathbf{X}_\star^{(\delta)}$ with $\delta = o(\varepsilon^3)$. Respectively denote the $(r, D)$-RSC and $(r, D)$-RSM constants of $\widehat{\mathcal{L}}^{\mathrm{full}}$ about $\mathbf{X}_\star$ as $\alpha = (\alpha^{(1)}, \ldots, \alpha^{(L)})$ and $\beta = (\beta^{(1)}, \ldots, \beta^{(L)})$. Assume $\alpha^{(l)} > 0$ and $\beta^{(l)} < \infty$ for each $1 \le l \le L$. Assume we use a LoRA module with rank $r \ge r_\star$. Then, every SOSP $(\mathbf{A}, \mathbf{B})$ of $\widehat{\mathcal{L}}_\lambda^{\mathrm{lora}}$ with $\mathbf{X}_\square = \mathbf{A}\mathbf{B}^\mathsf{T}$ and $\|\mathbf{X}_\square - \mathbf{X}_\star\|_F \le D$ satisfies the following.*

1. *If $2\alpha^{(l)} \ge \beta^{(l)}(1 + \varepsilon)$ for all $l = 1, \ldots, L$ (special regime), $\mathbf{X}_\square$ is an $\varepsilon$-global minimizer.*

2. *If $2\alpha^{(l)} < \beta^{(l)}(1 + \varepsilon)$ for some $l = 1, \ldots, L$ (generic regime), one of the following holds.*

   (i) *$\mathbf{X}_\square$ is an $\varepsilon$-global minimizer.*

   (ii) *$\mathbf{X}_\square$ is not an $\varepsilon$-global minimizer, $X_\square^{(l)}$ is exactly rank $r$ with $\sigma_r(X_\square^{(l)}) \ge \max\{\frac{2\alpha}{\beta(1+\varepsilon)}\sigma_{r_\star}(X_\square^{(l)}), \frac{\alpha}{2\beta\sqrt{r}} \cdot \varepsilon\}$, and either*

   $$\sigma_r(X_\square^{(l)}) \le \frac{2\alpha}{\beta}\sigma_{r_\star}(X_\square^{(l)})$$

   *or*

   $$\left\|X_\square^{(l)} - X_\star^{(l)}\right\|_F \ge \sqrt{\frac{\left\|X_\square^{(l)} - \Pi_{\mathrm{rank} \le r_\star}(X_\square^{(l)})\right\|_F^2 - \varepsilon^3}{1 - \frac{2\alpha\sigma_{r_\star}}{\beta\sigma_r}}} - \varepsilon^2$$

   *where $\Pi_{\mathrm{rank} \le r_\star}(X_\square^{(l)})$ is the projection of $X_\square^{(l)}$ onto the set of matrices of rank $r_\star$ or less.*

*To clarify, when we say $X_\square$ is or is not an $\varepsilon$-global minimizer, it is with respect to $\widehat{\mathcal{L}}_\lambda^{\mathrm{full}}$.*

*Proof.* We can proceed identically with the proof of Theorem 2 up to defining $Z$ such that $\|Z\|_2 \le \sigma_r$.

We first prove that $\sigma_r(X^{(l)}) \ge \varepsilon \cdot \frac{\alpha}{2\beta\sqrt{r}}$ if $\|X^{(l)} - X_\star^{(l)}\| \ge \varepsilon$, which requires independent reasoning, and then we prove the remaining results analogously to Theorem 1.

**Step 1.** Fix some constant $c > 0$, and define $r' = \min\{\gamma \,|\sigma_\gamma(X^{(l)}) < c, \; 1 \le \gamma \le r\}$. Now setting $\kappa$ as a positive real number that $\kappa \times \beta\sigma_r < c$. By the Eckart–Young–Mirsky theorem we see

$$\underset{\mathrm{rank}(Y) \le r_\star}{argmin} \|Y - (X^{(l)} - \kappa Z)\|_F^2 = (X^{(l)} - \kappa Z)^{r_\star}$$

where $(X^{(l)} - \kappa Z)^{r_\star}$ is the projection of $X^{(l)} - \kappa Z$ onto the set of matrices of rank $r_\star$ or less. Since $\|\kappa Z\|_2 < c$, by definition of $r'$ the 1-th to $r$-th singular vectors of $X^{(l)} - \kappa Z$ coincide with those of $X^{(l)}$, while the subsequent singular values are all smaller than $c$. This implies

$$\|(X^{(l)})^{r'} - (X^{(l)} - \kappa Z)^{r_\star}\|_F^2 \le c^2 \cdot \max\{r_\star - r', 0\}$$

If $r_\star < r'$, we are done, so we can consider only the case where $r_\star - r' \geq 0$. Combining the two relations, we have

$$\|(X^{(l)})^{r'} - (X^{(l)} - \kappa Z)\|_F \leq \|(X^{(l)})^{r'} - (X^{(l)} - \kappa Z)^{r_\star}\|_F + \|(X^{(l)} - \kappa Z)^{r_\star} - (X^{(l)} - \kappa Z)\|_F$$
$$\leq c\sqrt{r_\star - r'} + \|(X_\star^{(\delta)})^{(l)} - (X^{(l)} - \kappa Z)\|_F$$

and squaring each sides, this expands to

$$\|(X^{(l)})^{r'} - X^{(l)}\|_F^2 \leq c^2(r_\star - r') + 2c\sqrt{r_\star - r'}\|(X^{(l)})^{r'} - (X^{(l)} - \kappa Z)\|_F + \|(X_\star^{(\delta)})^{(l)} - X^{(l)}\|_F^2 + 2k\langle(X_\star^{(\delta)})^{(l)} - (X^{(l)})^{r'}, Z\rangle$$

Now in the same way with A.1, due to the restricted convexity and the subgradient property we have

$$2\alpha\kappa\|X^{(l)} - X_\star^{(l)}\|_F^2 \leq 2k\langle X^{(l)} - X_\star^{(l)}, Z\rangle.$$

Adding the two up, we have

$$(2\alpha\kappa - 1)\|X^{(l)} - X_\star^{(l)}\|_F^2 + \|(X^{(l)})^{r'} - X^{(l)}\|_F^2 \leq c^2(r_\star - r) + 2c\sqrt{r_\star - r'}\|(X^{(l)})^{r'} - (X^{(l)} - \kappa Z)\|_F$$
$$+ (\|(X_\star^{(\delta)})^{(l)} - X^{(l)}\|_F^2\|X_\star^{(l)} - X^{(l)}\|_F^2) + 2k\langle(X_\star^{(\delta)})^{(l)} - X_\star^{(l)}, Z\rangle + 2k\langle X^{(l)} - (X^{(l)})^{r'}, Z\rangle$$

which again simplifies to

$$(2\alpha\kappa - 1)\|X^{(l)} - X_\star^{(l)}\|_F^2 - 2\delta\|X_\star^{(l)} - X^{(l)}\|_F \leq 3c^2(r - r') + 2c\delta\sqrt{r - r_\star},$$

or equivalently if $2\alpha\kappa - 1 > 0$,

$$\|X^{(l)} - X_\star^{(l)}\|_F \leq \frac{\delta}{2\alpha\kappa - 1} + \sqrt{(\frac{\delta}{2\alpha\kappa - 1})^2 + 3c^2(r - r') + 2c\delta\sqrt{r - r_\star}}.$$

Therefore, setting $c = \frac{\varepsilon}{2\sqrt{r - r'}}$ and $\kappa = \frac{c}{\beta\sigma_r}$ (or any large number if $\sigma_r = 0$), if $\sigma_r(X^{(l)}) < \frac{\alpha c}{\beta}$, then $\delta = o(\varepsilon^3)$ implies $\|X^{(l)} - X_\star^{(l)}\|_F < \varepsilon$. Thus if $\|X^{(l)} - X_\star^{(l)}\|_F \geq \varepsilon$, then $\sigma_r(X^{(l)}) \geq \frac{\alpha}{2\beta\sqrt{r}} \cdot \varepsilon$ for some $l$.

**Step 2.** Similarly to Theorem 1, define $\kappa$ as a positive real number that $\kappa \cdot \beta\sigma_r < \sigma_{r_\star}$, implying $\|\kappa Z\| \leq \sigma_{r_\star}$. Then

$$(X^{(l)})^{r_\star} \in \operatorname*{argmin}_{\operatorname{rank}(Y) \leq r_\star} \|Y - (X^{(l)} - \kappa Z)\|_F^2$$

and $\operatorname{rank}((X_\star^{(\delta)})^{(l)}) = r_\star$ so

$$\|(X^{(l)})^{r_\star} - X^{(l)} + \kappa Z\|_F^2 \leq \|(X_\star^{(\delta)})^{(l)} - X^{(l)} + \kappa Z\|_F^2$$

which is equivalent to

$$\|(X^{(l)})^{r_\star} - X^{(l)}\|_F^2 \leq \|(X_\star^{(\delta)})^{(l)} - X^{(l)}\|_F^2 + 2k\langle(X_\star^{(\delta)})^{(l)} - X^{(l)}, Z\rangle.$$

Now the relations (4),(5) from the proof of Theorem 1 still apply here so adding these we have

$$\alpha\|X^{(l)} - (X_\star^{(\delta)})^{(l)}\|_F^2 \leq \kappa\langle X^{(l)} - X_\star^{(l)}, Z\rangle$$

so adding the two inequalities, we obtain

$$2\kappa\alpha\|X^{(l)} - (X_\star^{(\delta)})^{(l)}\|_F^2 - \|X^{(l)} - (X_\star^{(\delta)})^{(l)}\|_F^2 + \|(X^{(l)})^{r_\star} - X^{(l)}\|_F^2 \leq 2k\langle(X_\star^{(\delta)})^{(l)} - X_\star^{(l)}, Z\rangle.$$

Here we can upper bound $\langle(X_\star^{(\delta)})^{(l)} - X_\star^{(l)}, Z\rangle$ by $2\sigma_{r_\star} \cdot \|(X_\star^{(\delta)})^{(l)} - X_\star^{(l)}\|_*$, by the duality of the nuclear norm and spectral norm and $\|\kappa Z\|_2 \leq \sigma_{r_\star}$. $\sigma_{r_\star}$ is again bounded by $\frac{D}{\sqrt{r_\star}}$ since we are looking at $\|X^{(l)}\|_F \leq D$, and $\|(X_\star^{(\delta)})^{(l)} - X_\star^{(l)}\| \leq \sqrt{r - r_\star} \cdot \|(X_\star^{(\delta)})^{(l)} - X_\star^{(l)}\| \leq \delta \times \sqrt{r - r_\star}$ by the Cauchy-Schwartz inequality. Therefore we have

$$2\kappa\alpha\|X^{(l)} - (X_\star^{(\delta)})^{(l)}\|_F^2 - \|X^{(l)} - (X_\star^{(\delta)})^{(l)}\|_F^2 + \|(X^{(l)})^{r_\star} - X^{(l)}\|_F^2 \leq \delta D\sqrt{\frac{r - r_\star}{r_\star}}.$$

We can expand this inequality, using $X^{(l)} - (X_\star^{(\delta)})^{(l)} = (X^{(l)} - X_\star^{(l)}) + (X_\star^{(l)} - (X_\star^{(\delta)})^{(l)})$ as

$$(2\kappa\alpha - 1)\|X_\star^{(l)} - X^{(l)}\|_F^2 + \|(X^{(l)})^{r_\star} - X^{(l)}\|_F^2 \leq 2\langle X^{(l)} - X_\star^{(l)}, X_\star^{(l)} - (X_\star^{(\delta)})^{(l)}\rangle + \delta D\sqrt{\frac{r - r_\star}{r_\star}}$$

$$\leq 2\delta\|X^{(l)} - X_\star^{(l)}\|_F + \delta D\sqrt{\frac{r - r_\star}{r_\star}}.$$

Solving the quadratic inequality about $\|(X_\star^{(\delta)})^{(l)} - X^{(l)}\|_F$, if $2\kappa\alpha - 1 > \varepsilon$ the discriminant is

$$\delta^2 + 4(2\kappa\alpha - 1)\left(\delta D\sqrt{\frac{r - r_\star}{r_\star}} - \|(X^{(l)})^{r_\star} - X^{(l)}\|_F^2\right).$$

Now we know that if $\mathbf{X}$ is an $\varepsilon$-spurious local minima then $\|(X^{(l)})^{r_\star} - X^{(l)}\|_F \geq \sigma_r \geq \frac{\alpha}{2\beta\sqrt{r}} \cdot \varepsilon$ for some $l$, so the discriminant is negative since $\delta = o(\varepsilon^2)$ and therefore there would be no spurious local minima. If $2\kappa\alpha - 1 < 0$, we would see

$$\|X_\star^{(l)} - X^{(l)}\|_F \leq \sqrt{\frac{\|(X^{(l)})^{r_\star} - X^{(l)}\|_F^2 - (\delta D\sqrt{\frac{r - r_\star}{r_\star}} - \frac{\delta^2}{1 - 2\kappa\alpha})}{1 - 2\kappa\alpha}} - \frac{\delta}{1 - 2\kappa\alpha}$$

resulting in

$$\|X_\star^{(l)} - X^{(l)}\|_F \leq \sqrt{\frac{\|(X^{(l)})^{r_\star} - X^{(l)}\|_F^2 - \varepsilon^3}{1 - 2\kappa\alpha}} - \varepsilon^2$$

given $\delta = o(\varepsilon^3)$ $\qquad\qquad\qquad\qquad\qquad\qquad\qquad\qquad\qquad\qquad\qquad\qquad\qquad\square$

## A.4. Proof of Lemma 1

*Proof.* Denote the low rank matrices at the $t$th iteration as $A_t \in \mathbb{R}^{m \times r}$ and $B_t \in \mathbb{R}^{n \times r}$, the weight update $X_t = A_t B_t^\intercal$, and the mini-batch used at the $t$th iteration as $S_t$. With a slight abuse of notation, denote the mini-batch loss for batch $S_t$ at iteration $t$ as

$$\widehat{\mathcal{L}}_{S_t}(X_t) = \frac{1}{b}\sum_{(x,y)\in S_t}\ell(f(W_0 + X; x), y).$$

Denote the input vector to the layer as $u \in \mathbb{R}^{m \times 1}$, where $u^\intercal \cdot (W_0 + AB^\intercal)$ is inputted to the next layer. Here we see that

$$\nabla_A\widehat{\mathcal{L}}_{S_t}(X_t) = \frac{1}{b}\sum_{(x,y)\in S_t}\nabla_A\ell(f(W_0 + AB^\intercal; x), y).$$

Now the gradient of the loss is expandable by the chain rule as

$$\nabla_A\ell(f(W_0 + AB^\intercal; x), y) = \frac{\partial\ell(f(W_0 + AB^\intercal; x), y)}{\partial A} = \frac{\partial\ell(f, y)}{\partial f} \cdot \frac{\partial f(W_0 + AB^\intercal; x)}{\partial A} = \frac{\partial\ell(f, y)}{\partial f} \cdot u \cdot \left(\frac{\partial f}{\partial(u^\intercal A)}\right)^\intercal.$$

Therefore $\nabla_A\ell(f(W_0 + AB^\intercal; x), y)$ is a rank-1 matrix, as it is a product of a scalar and two rank-1 matrices. Thus, $\widehat{\mathcal{L}}_{S_t}(X_t)$, as a sum of $b$ rank-1 matrices, has rank at most $b$.

Now we can characterize the SGD process with weight decay as

$$A_{t+1} = A_t - \mu\nabla_A\widehat{\mathcal{L}}_{S_t}(X_t) - 2\mu\lambda A_t$$
$$B_{t+1} = B_t - \mu\nabla_B\widehat{\mathcal{L}}_{S_t}(X_t) - 2\mu\lambda B_t$$

or equivalently,

$$A_{t+1} = (1 - 2\mu\lambda)A_t - \mu\nabla_A\widehat{\mathcal{L}}_{S_t}(X_t)$$
$$B_{t+1} = (1 - 2\mu\lambda)B_t - \mu\nabla_B\widehat{\mathcal{L}}_{S_t}(X_t).$$

Recursively applying this process $n$ times, we have

$$A_t = (1 - 2\mu\lambda)^n A_{t-n} - \mu \sum_{j=1}^{n} (1 - 2\mu\lambda)^{j-1} \widehat{\mathcal{L}}_{S_{t-j}}(X_{t-j}).$$

Therefore denoting $U_{t,n} := -\mu \sum_{j=1}^{n} (1 - 2\mu\lambda)^{j-1} \widehat{\mathcal{L}}_{S_{t-j}}(X_{t-j})$ we see $A_t, B_t$ are approximately close to $U_{t,n}, V_{t,n}$ by

$$\|A_t - U_{t,n}\| \leq (1 - 2\mu\lambda)^n \|A_{t-n}\|, \quad \|B_t - V_{t,n}\| \leq (1 - 2\mu\lambda)^n \|B_{t-n}\|$$

for analogously defined $V_{t,n}$, and $\mathrm{rank}(U_{t,n}), \, rank(V_{t,n}) \leq nb$. Therefore,

$$X_t = A_t B_t^\intercal = (U_{t,n} + (1 - 2\mu\lambda)^n A_{t-n})(V_{t,n}^\intercal + (1 - 2\mu\lambda)^n B_{t,n}^\intercal)$$
$$= U_{t,n}(V_{t,n}^\intercal + (1 - 2\mu\lambda)^n B_{t-n}^\intercal) + (1 - 2\mu\lambda)^n A_{t-n} V_{t,n}^\intercal + (1 - 2\mu\lambda)^{2n} A_{t-n} B_{t-n}^\intercal$$

Since $U_{t,n}(V_{t,n}^\intercal + (1 - 2\mu\lambda)^n B_{t-n}^\intercal)$, $(1 - 2\mu\lambda)^n A_{t-n} V_{t,n}^\intercal$ are both matrices of rank at most $b$, we can define a matrix $W$ with

$$\left\| \frac{X_t}{\|X_t\|} - W \right\| = (1 - 2\mu\lambda)^{2n} \frac{\|X_{t-n}\|}{\|X_t\|}$$

and $\mathrm{rank}(W) \leq 2nb$. Since we assumed $X_t$ converges to $\tilde{X}$, when $t$ is sufficiently large we can assume $\frac{\|X_{t-n}\|}{\|X_t\|} \leq 2$. Thus for any $n$ that $(1 - 2\mu\lambda)^{2n} < \varepsilon/2$, there exists some matrix $W$ with

$$\left\| \frac{X_t}{\|X_t\|} - W \right\| < \varepsilon, \quad \mathrm{rank}(W) \leq b \times \frac{\log(\varepsilon/2)}{\log(1 - 2\mu\lambda)}$$

$\square$

## B. Matrix lemmas

**Lemma B.1.** *If matrices $A \in \mathbb{R}^{m \times r}, B \in \mathbb{R}^{n \times r}$ satisfy $A^\intercal A = B^\intercal B$,*

$$A = U\Sigma^{1/2} W^\intercal, \quad B = V\Sigma^{1/2} W^\intercal$$

*where $U\Sigma V^\intercal$ is a singular value decomposition of $AB^\intercal$, and $W$ is a orthonormal matrix.*

*Proof.* Take the compact singular value decompositions of $A$ and $B$,

$$A = U_A \Sigma_A V_A^\intercal, \quad B = U_B \Sigma_B V_B^\intercal.$$

Then we have

$$V_A \Sigma_A^2 V_A^\intercal = A^\intercal A = B^\intercal B = V_B \Sigma_B^2 V_B^\intercal$$

Here $V_A, V_B$ are both orthogonal matrices and $\Sigma_A^2, \Sigma_B^2$ are both diagonal matrices, so by the uniqueness of the singular value decomposition, by sufficient reordering and sign flipping of the singular vectors we can have $V_A = V_B, \Sigma_A = \Sigma_B$. Therefore inputting this into $AB^\intercal = X^{(l)}$, we have

$$U_X^{(l)} \Sigma_X V_X^{(l)\intercal} = U_A \Sigma_A^2 U_B^\intercal$$

which implies again $\Sigma_A = \Sigma_X^{1/2}, U_A = U_X^{(l)}, U_B = V_X^{(l)}$ up to reordering and sign flipping. This proves the given statement.

$\square$

**Lemma B.2.** *For orthonormal matrices $U \in \mathbb{R}^{m \times r}$ and $V \in \mathbb{R}^{n \times r}$, if $X \in \mathbb{R}^{m \times n}$ satisfies $XV = U$ and $X^\intercal U = V$ then $X$ can be expressed as*

$$X = UV^\intercal + \tilde{U}\Sigma\tilde{V}^\intercal$$

*where $\tilde{U} \in \mathbb{R}^{m \times (m-r)}$ is an orthogonal basis for the orthogonal complement of the column space of $U$, $\tilde{V} \in \mathbb{R}^{n \times (n-r)}$ is an orthogonal basis for the orthogonal complement of the column space of $V$, and $\Sigma \in \mathbb{R}^{(m-r) \times (n-r)}$ is a rectangular diagonal matrix.*

*Proof.* First fix an arbitrary $\tilde{U}$ and $\tilde{V}$, that makes $[U\ \tilde{U}]$ and $[V\ \tilde{V}]$ orthogonal. Since orthogonal matrices are invertible, $X$ can be expressed as

$$X = [U\ \tilde{U}] \begin{bmatrix} A & B \\ C & D \end{bmatrix} \begin{bmatrix} V^\intercal \\ \tilde{V}^\intercal \end{bmatrix}$$

for some $A \in \mathbb{R}^{r\times r}, B \in \mathbb{R}^{r\times(n-r)}, C \in \mathbb{R}^{(m-r)\times r}, D \in \mathbb{R}^{(m-r)\times(n-r)}$. Expanding this expression we can write $X$ again as

$$X = UAV^\intercal + UB\tilde{V}^\intercal + \tilde{U}CV^\intercal + \tilde{U}D\tilde{V}^\intercal$$

Therefore $XV = U, X^\intercal U = V$ implies $UA + \tilde{U}C = U, VA + \tilde{V}B = U$, respectively. Now the orthogonality of $U, \tilde{U}$ and $V, \tilde{V}$ implies $A = I_r$ and $B, C = 0$. Finally, let the singular value decomposition of $D$ be $D = U_D\Sigma V_D^\intercal$. Then

$$A = UV^\intercal + \tilde{U}U_D\Sigma_D(\tilde{V}V_D)^\intercal$$

and $\tilde{U}U_D, \tilde{V}V_D$ are still orthonormal matrices in the same space, so our statement is proved.

$\square$

# C. Experimental Details

## C.1. Verifying low-rank global minima exist

**Experimental setup**   We perform the experiments by fine tuning a RoBERTa model for the SST-2 dataset, and fine tuning a Vision Transformer model for the CIFAR-100 dataset. To maintain only low rank updates, we follow Malladi et al. (2023), by first training the linear classification head only, and fine tuning on the resulting model. This way of training is in fact quite reasonable, as prior studies report it is beneficial to LP-LoRA (Tomihari & Sato, 2024), i.e. perform linear probing first and then apply low rank adaptation. We tune a total of 24 matrices for each experiment, with the hyperparameters as below.

Table 3: Hyperparameters for verifying low-rank global minima exist

| HyperParameter | SST2 | CIFAR100 |
|---|---|---|
| Task | SST2 | CIFAR100 |
| $\lambda$ | 0, 0.1, 0.05, 0.01, 0.005, 0.001 | 0, 0.01, 0.005, 0.003, 0.001, 0.0005 |
| Learning Rate | 0.005 | 0.001 |
| Scheduler | Cosine Annealing | Cosine Annealing |
| Optimizer | Proximal SGD | Proximal SGD |
| Batch Size | 128 | 128 |
| Epochs | 150 | 150 |

**Optimizing nuclear norm**   For verifying the existence of a low rank global minima exists, we optimize over the loss $\widehat{\mathcal{L}}_\lambda^{full}(X) = \widehat{\mathcal{L}}(X) + \lambda\|X\|_\star$. Here the nuclear norm is non-differentiable, so instead of the standard stochastic gradient descent, we apply proximal gradient method, well known to converge to a global minimum in a convex objective (Parikh & Boyd, 2014):

$$X_{t+1} = \text{prox}_{\mu\lambda\|\cdot\|_*}(X_t - \mu\nabla\widehat{\mathcal{L}}(X_t))$$

where

$$\text{prox}_{\mu\lambda\|\cdot\|_*}(X) = \underset{X'}{\text{argmin}} \left( \frac{1}{2\mu}\|X' - X\|_F^2 + \lambda\|X'\|_* \right).$$

**Computing the rank**   To effectively compute the rank considering numerical issues, we use the torch.linalg.svd function to obtain the singular values of the normalized weight matrix, and truncate values below $10^{-4}$. As we consider approximately low rank matrices in Section 3.3, our theory robustly holds for this truncated approximate rank as well.

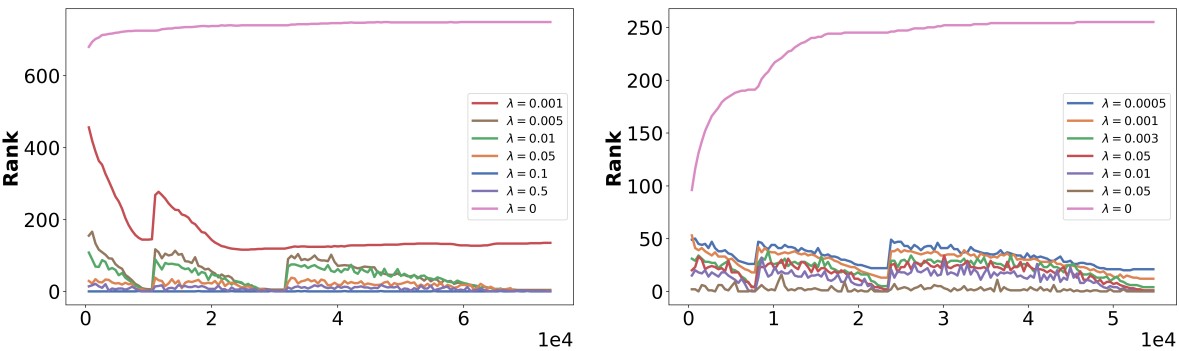

Figure 3: Rank of the first value matrix throughout training. (left) SST2, (right) CIFAR100

**Results**  The final ranks after training are presented well in Table 1. Here we present the change of the ranks throughout training. For simplicity, we present the rank of the first value matrix, which had the largest rank among all matrices in general. The results presented in Table 1 are the ranks of the weight matrix with the highest rank for each $\lambda$.

We also present the test accuracy curves for each task. We see that weight decay have contrastive impacts for sst2 and cifar100, where weight decay doesn't harm the generalization performance as long as it is not too big in SST2, while it significantly impacts performance for CIFAR100. Therefore we do not use CIFAR100 for the following experiments, as the weight-decay equipped solutions are not the ones of our interest. Nevertheless, we clearly see the decreasing trend of rank as a function of $\lambda$, therefore successfully verifying our assumption.

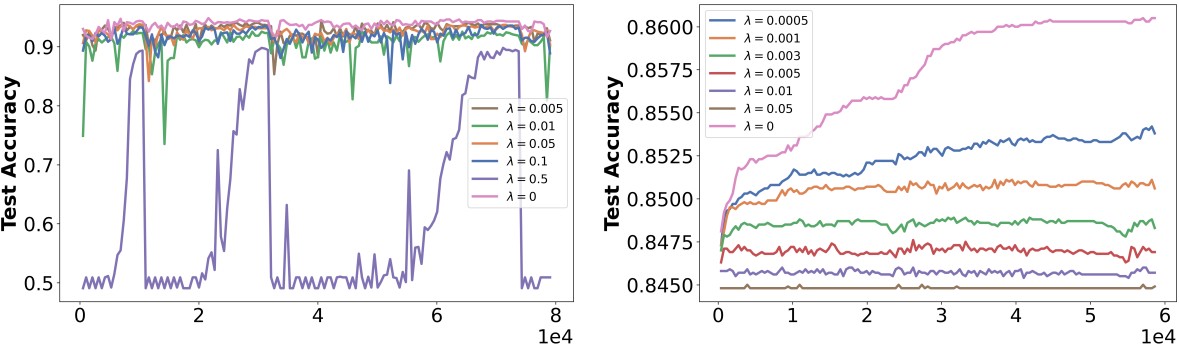

Figure 4: Test accuracy throughout training. (left) SST2, (right) CIFAR100

## C.2. Verifying RSC and RSM

Precisely computing the RSC and RSM constants of a deep neural network is generally infeasible, and therefore we perform monte-carlo sampling for 1000 samples to find an estimate for these values. Specifically, we compute

$$\alpha^{(l)} = \min_{1 \leq i \leq N} \frac{\langle \nabla f_l(\mathbf{X_i}) - \nabla f_l(\mathbf{X_\star}), X_i - X_\star^{(l)} \rangle}{\|X_i^{(l)} - X_\star^{(l)}\|_F^2}$$

$$\beta^{(l)} = \max_{1 \leq i \leq n} \left\{ \max_{\substack{\|U\|_F = \|V\|_F = 1 \\ \mathrm{rank}(U) = \mathrm{rank}(V) = 1}} \frac{\nabla_{l,l}^2 f(\mathbf{X})[UX^{(l)} + X^{(l)}V, UX^{(l)} + X^{(l)}V]}{\|UX^{(l)} + X^{(l)}V\|_F^2} \right\}$$

for samples $\mathbf{X}_1, \ldots, \mathbf{X}_N$ with $\|X_i^{(l)} - X_\star^{(l)}\| \leq D$, $\mathrm{rank}(X_i^{(l)}) \leq r$ for each $1 \leq i \leq N$. Due to the intense computational bottleneck of the Hessian, we limit the computation to only the last layer of $W_q$ and $W_v$, following (Jang et al., 2024). The $\alpha$, $\beta$ values we present in Section 4 are the values for $W_v$, which had larger $\frac{\beta}{\alpha}$ values than $W_q$ for each case.

## C.3. Validating main theorem

In Section 4, we present the training results in two cases: zero initialization and Large initialization, which demonstrate a global minimizer and spurious local minimizer, respectively. We additionally present here that in smaller initializations, and in other setups, LoRA training always converges to the global minimum, indeed validating our argument that spurious local minima *must* be unfeasible if they exist.

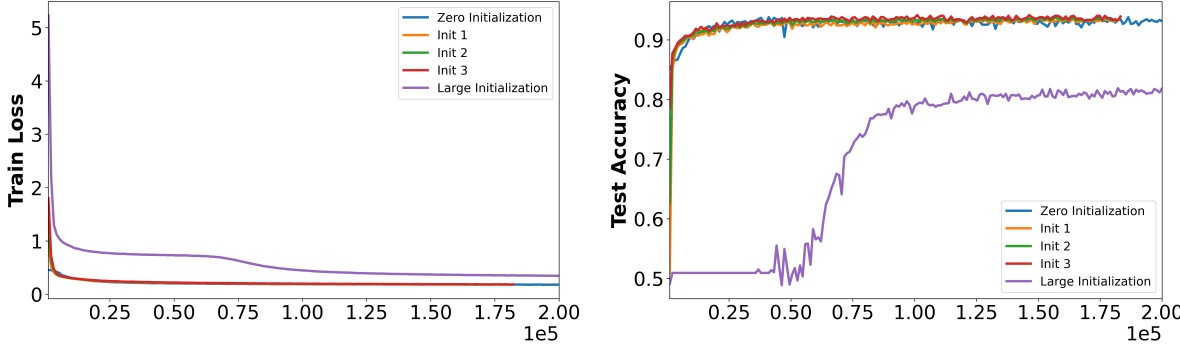

Figure 5: LoRA training on SST2 with varying initialization (left) training loss, (right) test accuracy

We present the specific initializations below. Here the initialization for the $A$ matrix in the zero init follows the standard Kaiming uniform initialization, while other initializations are random gaussian initializations.

| Matrix | A | B |
|---|---|---|
| Zero Initialization | $\mathcal{U}(-\sqrt{\frac{15}{768}}, \sqrt{\frac{15}{768}})$ | $0$ |
| Initialization 1 | $\mathcal{N}\left(0, \frac{1}{30}\right)$ | $\mathcal{N}\left(0, \frac{1}{30}\right)$ |
| Initialization 2 | $\mathcal{N}\left(0, \frac{1}{10}\right)$ | $\mathcal{N}\left(0, \frac{1}{10}\right)$ |
| Initialization 3 | $\mathcal{N}\left(-\frac{1}{10}, \frac{1}{20}\right)$ | $\mathcal{N}\left(\frac{1}{10}, \frac{1}{20}\right)$ |
| Large Initialization | $\mathcal{N}\left(0, \frac{1}{5}\right)$ | $\mathcal{N}\left(0, \frac{1}{5}\right)$ |

Table 4: Initialization values for matrices A and B

