# OpenReview forum: "LoRA Training Provably Converges to a Low-Rank Global Minimum Or It Fails Loudly (But it Probably Won't Fail)"
_ICML.cc/2025/Conference — ICML 2025 oral_

### Official Review · Reviewer_Rhr4 · 2025-03-10

**Overall Recommendation:** 3

**Summary:**

The authors investigate the landscape of LoRA fine-tuning under assumptions of restricted strong convexity and smoothness. In particular, the authors prove a characterization of second-order stationary points for problems with regularization, showing that spurious high-rank local minima are bounded away from the global minimizers.

**Claims And Evidence:**

The claims of the authors sound reasonable and the proofs look correct under the authors' assumption. However, I fail to recognize the quantitative part of the result, in fact Theorem 1 does not exclude the case of numerical low-rank spurious local minima. See "questions" for more details.

**Essential References Not Discussed:**

As far as I am concerned, all the relevant literature was discussed.

**Experimental Designs Or Analyses:**

The experimental evaluation is not satisfying in relation to the theoretical claims. I would have appreciated a more controlled numerical setting (e.g. we know the global minima) to show convergence and the possible absence of spurious local minima around.

**Methods And Evaluation Criteria:**

I believe the experimental evaluation lacks both a controlled setting to show quantitatively the authors' claims.

**Other Comments Or Suggestions:**

See "Experimental Designs Or Analyses".

**Other Strengths And Weaknesses:**

The results in the paper are sound and shed light on why zero initialization works well in practice for LoRA fine-tuning.

**Questions For Authors:**

While qualitatively the result in theorem 1 and corollary 1 state that there might be spurious local minimizes of higher rank far from the global one, it doesn't eliminate the possibility of spurious local minima with numerical low-rank (therefore, potentially close to $X_*$. Can the authors comment on this?

**Relation To Broader Scientific Literature:**

LoRA variants are abundant in literature, and their popularity exploded in the last years given their effectiveness. The authors' work focuses on showing that the landscape of these problems is in some way "well-behaving", with low-rank local minima and higher rank spurious minima which are bounded away from the global one.

**Theoretical Claims:**

The theoretical claims and the proofs provided by the authors are correct as far as I am concerned.

---

> ### Author Rebuttal · Authors · 2025-04-01
>
> ### **Common Response (Repeated in all responses)**
>
> First of all, we thank the reviewers for their positive and constructive feedback. We are excited to see that the reviewers are appreciative of our theoretical contributions. Below, we address each of the reviewers' comments individually.
>
> ### **Individual Response**
>
> We are happy to hear that the reviewer found our results sound and illuminating. Below, we address the reviewer’s main concern regarding the scenario with near-low-rank updates.
>
>
> **Q) Theorem 1 and Corollary 1 … spurious local minimizes of higher rank far from the global one, it doesn't eliminate possibility of spurious local minima with numerical low-rank**
>
> We thank the reviewer for the thoughtful question. We clarify the claims of Theorem 1 and Corollary 1 as follows.
>
> Corollary 1 states that if $X_\square$ is a spurious local minima, then $\sigma_r (X_\square) \ge \frac{2\alpha}{\beta} \sigma_{r*} (X_\square)$. This provides a substantive lower bound on the $r$th singular value, making $X_\square$ not numerically low rank. (Recall that $r> r_\star$ in Corollary 1 and “low-rank” refers to rank $r_\star$.) Moreover, this lower bound on $\sigma_r (X_\square)$ means that $X_\square$ and the global minimum $X_\star$ cannot be close, since $\sigma_r(X_\star)=0$. Therefore, Corollary 1 does eliminate the possibility that a spurious local minimum is low-rank or is close to $X_\star$.

---

> > ### Comment · Reviewer_Rhr4 · 2025-04-02
> >
> > First of all, I would like to thank the reviewers for the rebuttal.
> > I agree with your comment, but in practical problems, it is however possible to be in the ill-conditioned setting in which $\frac{\alpha}{\beta} \approx 0$. In particular, if $\frac{\alpha}{\beta} \approx 0$ was of order machine precision, it could be possible to find spurious local minima $X_\Box$ close to $X_*$ (as shown also in table2).
> > I believe for this reason it would be beneficial to include in the revised version an experiment in a controlled scenario, in which $\alpha$, $\beta$, and the global minima are controlled a priori (could be even a simple quadratic problem).
> > In this setting, it should be possible to showcase the distance to minima over time and your predictions concerning the stationary point. Moreover, in an experimental setting similar to this one one could also showcase the dependency of $||X_\Box -X_* ||$ on the condition number $\kappa = \beta/\alpha$, which is what I am personally interested in cause it could be crucial for applications.
> >
> > In any case, I am satisfied by the author's rebuttal and I am keeping my score.

---

### Official Review · Reviewer_9M1X · 2025-03-16

**Overall Recommendation:** 3

**Summary:**

This paper provides a theoretical analysis of Low-Rank Adaptation (LoRA)  loss landscape (near the global or local min).
The main contributions are as follows:

(1) The authors identify two regimes—special (well conditioned) and generic (more realistic). In the generic regime, second-order stationary points (SOSPs) are either (a) low-rank, small-magnitude global minima or (b) high-rank, large-magnitude spurious minima.

(2) The authors shows that zero-initialization and weight decay are biased toward low-rank, small-magnitude solutions, which partially explains LoRA's empirical success despite non-convexity.

(3) The paper provides some experimental results confirming the existence of low-rank global minima under nuclear norm regularization and validate restricted strong convexity/smoothness assumptions. They also illustrate that the failure cases (high-rank solution) may results in from large initialization.

**Claims And Evidence:**

The claims are supported by the theoretical proofs and experimental validation.

**Essential References Not Discussed:**

There are a flurry of recent works on improving the vanilla LoRA (by tuning the learning rate or changing the initialization etc.), such as LoRA+, rsLoRA, LoRA-GA, PiSSA. Does these work change the conclusion of the paper? Some papers propose to change the initialization of LoRA (e.g.,  LoRA-GA, PiSSA). Do these results contradict the result on zero-initialization?

**Experimental Designs Or Analyses:**

The main contribution of the paper is theoretical and the experimental section is relatively small.
I checked the experimental setting and the results, which look reasonable to me.

**Methods And Evaluation Criteria:**

The methods makes sense.

**Other Comments Or Suggestions:**

1. page 3. section 1.3 ||X||_\star, is this the nuclear norm? but the sentence before this line says "adding l_2 regularization" which is confusing.

2. page 3. section 1.3 line 129. the equivalence between the unconstrained minimization problem and the constrained minimization problem. Are they always equivalent? Is there any condition or assumption?

3. Information theorem line 153. The algorithm used in Ge et al. 2015 is somewhat different from the standard SGD. Moreover, the standard algorithm for training LoRA is Adam. So there is a discrepancy between the theorem cited and the conclusion (line 160) drawn from it.

typo:
(1) page 5, line 251
(2) page 11. line 571.

**Other Strengths And Weaknesses:**

Strengths:
(1) Theoretical Contributions: The paper provides a rigorous theoretical analysis of LoRA's loss landscape. The helps explain the success of LoRA in practice.
(2) The new theoretical analysis doesn't rely on linearization arguments.
Weaknesses:
(1) Limited Experimental Section: The experimental validation, while sufficient to support the theoretical claims, is relatively small in scope. It focuses on fairly small-scale specific tasks (SST-2 and CIFAR-100).
(2) Some assumptions and proofs requires clarification.
(3) Several recent important improvements on LoRA are not mentioned at all.

**Questions For Authors:**

I have asked questions in previous sections.

**Relation To Broader Scientific Literature:**

LoRA is part of a broader trend in machine learning towards developing parameter-efficient methods for adapting large pre-trained models. The contribution of the paper is useful in understanding the loss landscape (near the global/local min) of LoRA optimization.

The paper also discusses the implicit bias of zero-initialization and weight decay. This connects to a growing body of work on the implicit regularization effects of optimization algorithms like SGD in deep learning models.

**Theoretical Claims:**

The main theoretical results (Theorem 1 and Cor 1) are interesting. I checked the proofs (not every details) and most parts look correct to me. I have not identified any serious flaws, but some claims and notations requires some clarification.

(1) Why the resticted smoothness is defined in this form (it is somewhat different from the standard smoothness assumption). In particular, why U and V  appear here. More explanations are needed.
(2) In theorem 1, 2(ii), why X_box can be full rank? (since X_box is defined to be AB^T)
(3) Proof 1. In the proof, the authors used kappa to distinguish several cases of the theorem. However, it is unclear to me how the condition of kappa corresponds to the condition of alpha and beta. In particular, which case is the special regime and which case is the generic regime (this is not mentioned in the proof at all)?

---

> ### Author Rebuttal · Authors · 2025-04-01
>
> ### **Common Response (Repeated in all responses)**
>
> First of all, we thank the reviewers for their positive and constructive feedback. We are excited to see that the reviewers are appreciative of our theoretical contributions. Below, we address each of the reviewers' comments individually.
>
> ### **Individual Response**
> We are excited to hear that the reviewer found our main theoretical results interesting. We are also grateful for the detailed feedback provided by the reviewer, and we respond to them individually in the following.
>
> **Regarding the relation with other LoRA variants**
>
> Yes, as the reviewer points out, our theory is indeed applicable to the more modern enhanced LoRA variants. We discuss a few popular LoRA variants, including the ones pointed out by the reviewer.
>
> LoRA+ (Hayou et.al., 2024) and rsLoRA (Kalajdzievski, 2023) share LoRA's objective and initialization while modifying hyperparameters like the learning rate and scaling factors to accelerate and stabilize training. Therefore, our theory assures these variants can also successfully find the global minimum, hopefully, faster and more stable.
>
> PiSSA (Meng et.al., 2024) and MiLoRA (Wang et.al., 2024) modify the training objective by decomposing the pretrained weight matrix into the sum of a principal ($A_pB_p$) and minor($A_mB_m$) component. PiSSA optimizes the objective $f(A_{lora}B_{lora} +A_mB_m)$, initializing $A_{lora}B_{lora}$ with $A_pB_p$. In contrast, MiLoRA optimizes $f(A_pB_p+A_{lora}B_{lora})$, initializing with $A_mB_m$. While appearing very similar, our theory introduces a contrasting view for the two methods.
>
> PiSSA initializes the fine-tuning weights with $A_pB_p$, which inherently contains 'principal' pretrained model information. In this setting, applying weight decay to $A_p$ and $B_p$ is counterintuitive; rather than regularizing the fine-tuned model to remain close to the pretrained model, weight decay here would bias the model toward losing essential pretrained characteristics. Thus a low-rank solution isn't naturally expected, and the model would be required to be very well-conditioned (i.e., within the special regime) to guarantee convergence to the global minimum under our framework.
>
> Conversely, the use of weight decay in MiLoRA is more sensible as $A_mB_m$ inherently represents a minor component. Therefore, we can more naturally apply our results to MiLoRA to guarantee convergence to a global minimum, in contrast with PiSSA. Investigating the practical implications of this theoretical insight is an interesting direction of future work.
>
> We note that variants like LoRA-GA (Wang et.al., 2024), which significantly alter training dynamics, do fall outside our current scope and offer promising directions for future work.
>
> **Q) Why the restricted smoothness …**
>
> Our definitions of restricted convexity and smoothness are motivated by the proofs requiring only a "directional form" of strong convexity and smoothness. In contrast, the conventional definitions require strong convexity and smoothness for all possible directions. While our definitions may feel more complicated, they are weaker and more realistic assumptions compared to standard strong convexity and smoothness. In particular, instead of requiring the standard smoothness $\nabla^2 f(X) [A, A] \le \beta ||A||_F^2 $ for all $A \in \mathbb{R}^{m\times n}$, we only require it for the directions$A$ in the space $\\{UX+XV | rank(U)=rank(V)=1 \\}$, a much smaller, restricted space.
>
> **Q) Why X_box can be full rank**
>
> We apologize for this confusion. By “full rank” meant rank $r$, since the low-rank update $X_{box}=AB^T$ can have rank at most $r$. However, we now see that this was not the correct language, so we will update it to say rank $r$ instead of full rank.
>
> **Q) In Proof 1, the authors used kappa …**
>
> As in line 253, $\kappa$ is defined as $\frac{\sigma_{r*}}{\beta \sigma_r}$. Thus, the case $2\kappa \alpha >1$ in the last line of the proof corresponds to the special regime and case (i) of the generic regime, while the converse corresponds to the case (ii) of the generic regime. Thank you for pointing this out, we will make this distinction clearer in our revision.
>
> **Q) ||X||_\star, is this the nuclear norm? but … Q) equivalence between the unconstrained…**
>
> We apologize for the confusion. The L2 regularization applies to the LoRA fine-tuning objective, while the nuclear norm regularization applies to the rank-constrained full fine-tuning objective. As we note in Section 1.3, these two objectives are equivalent, as shown in Recht et.al., 2010, Lemma 5.1.
>
> **Q) Ge et al. 2015 is somewhat different from the standard SGD … Adam.**
>
> We appreciate the reviewer’s precision on this point. While we believe that the qualitative point we derive through this reference remains valid, we agree with the reviewer that, strictly speaking, Ge et al.’s result does not imply global convergence for all optimizers. We will clarify this point explicitly in our revised paper.

---

### Official Review · Reviewer_wCMH · 2025-03-16

**Overall Recommendation:** 4

**Summary:**

This paper provides a theoretical understanding of the training dynamics of LoRA (i.e. Low-Rank Adaptation) of transformers. The authors first establish the equivalence between the low-rank form of loss and the rank-constrained optimization problem. Then the authors state their main results that the LoRA results can be split into two regimes, and in the second regime ("generic regime") LoRA can lead to "failed" results (i.e., does not converge to a global minimum and stuck in a spurious local minimum). The authors further demonstrate that standard LoRA practices, such as zero initialization and weight decay, and extend to multiple matrices fine-tuning.

**Claims And Evidence:**

I think the claims made in the submission are well supported by both theory and experiments.

**Essential References Not Discussed:**

N/A

**Experimental Designs Or Analyses:**

Yes. In my opinion the experiments only account for a very small part of this paper, and are mainly served as a validation of theory results. The authors come up with two settings and demonstrate the results fit the expected outcomes.

**Methods And Evaluation Criteria:**

The authors consider a few cases (both NLP and CV tasks) and conduct experiments to verify their main theorems. More specifically, the authors use two different initialization methods, one of which lead to global minimum and another converges to spurious local minimum. However, it would be helpful in my opinion to add another experiment to show that the multiple experiments can converge to the same global minimum.

**Other Comments Or Suggestions:**

There are some occasional misuse of \cite and \citep. It would be great if the authors can fix them. Example: "following the prescription of (Hu et al.,2022)." at line 378.

**Other Strengths And Weaknesses:**

This paper could be useful to identify / design better low-rank adaptation algorithms since it provides a principled way to identify if the fine-tuning could work or not. It can also be used as a monitor to tell if the training / finetuning is healthy or not.

**Questions For Authors:**

- It would be helpful in my opinion to add another experiment to show that the multiple experiments can converge to the same global minimum. Correct me if I am wrong but if the claims hold, should the solutions of low rank matrices be similar?

**Relation To Broader Scientific Literature:**

I am not familiar with relevant literature, but I checked the referenced works and I believe the authors' work has distinct aspects compared to those existing ones.

**Theoretical Claims:**

Although I have not got solid theoretical background, I tried my best to check the theoretical claims. Did not identify issues.

---

> ### Author Rebuttal · Authors · 2025-04-01
>
> ### **Common Response (Repeated in all responses)**
>
> First of all, we thank the reviewers for their positive and constructive feedback. We are excited to see that the reviewers are appreciative of our theoretical contributions. Below, we address each of the reviewers' comments individually.
>
> ### **Individual Response**
> We are happy to hear that the reviewer found our claims to be well supported by theory and experiments and the contribution distinct from prior works. We agree with the reviewer's view that “this paper could be useful to identify/design better low-rank adaptation algorithms,” and we outline how our theory can be applied to several LoRA variants in our response to Reviewer 9M1X.
>
> **Regarding the question of whether multiple experiments share the same global minimum**
>
> We thank the reviewer for the insightful observation and suggestion. Yes, our theorem indeed implies that multiple experiments should converge to the same global minimum (the same product $X=AB^\intercal$, but the low rank factors $A$ and $B$ are determined only up to rotations). To demonstrate this empirically, we extended the experiments originally presented in Figure 5 of the appendix to directly test whether training trajectories with multiple random seeds converge to the same limit. See the figure of the link below:
>
> https://drive.google.com/file/d/1vU2oZT2qAHxcY2gnIjA5CIBUwevxNJhs/view?usp=sharing
>
> In the figure, we plot the ‘total variation' of the training trajectories with multiple random seeds, defined as: $\sum_{1\le i<j\le N}  \|| \Theta_i^{(t)} - \Theta_j^{(t)} ||$ where $\Theta_i^{(t)}$ is the parameters of the $i$ th model at the end of the $t$th epoch. We see that the total variation converges to 0, indicating that multiple experiments share the same global minimum. (The total variation starts at 0 because the $B$ matrix is always initialized to be 0, so the product $X=AB^\intercal$ starts out at the same value of 0, even though $A$ is initialized differently.)

---

### Official Review · Reviewer_ni32 · 2025-03-24

**Overall Recommendation:** 3

**Summary:**

The authors have shown that a low rank and low magnitude initialisation in LoRA models results in convergence towards a global minima. Conversely, larger rank models with larger initialisation variance results in convergence towards spurious local minima with high probability. Whilst this is a result observed in experimentation in previous literature, this is largely a novel result for theoretical confirmation.

# update after rebuttal The rebuttal discussion was informative and helped to understand some aspects of this paper better. I will maintain my original score

**Claims And Evidence:**

As mentioned in the summary, there is a theoretical claim that low rank low magnitude initialisation results in convergence towards global minima. This is supported by a mathematical proof. Some experimental evidence is present.
The results in Table 2 maybe even understated – an interesting insight which both highlights and limits the application of this theory to LoRA models (that convexity assumption $\alpha$>0 does not hold for larger rank).
Contrasting over parameterisation is very interesting, could be worth discussing that more prominently.

**Essential References Not Discussed:**

None noted

**Experimental Designs Or Analyses:**

Experimental design does seem to be a bit 'rushed' but does validates and supports mathematical proofs. Table 2 results are of interest, maybe worth discussing more.

**Methods And Evaluation Criteria:**

Use of CIFAR 100 is a reasonably complex dataset for showing differences in initialisations during low-rank adapted training. The paper presents results for sufficient number of steps and there is a good use of metrics like rank (which was particularly novel for readers/reviewers) for showing convergence differences.

**Other Comments Or Suggestions:**

* Page 2, line 76, column 2: “for any any matrix” -- “for any matrix”
* Page 3, line 152, column 2: “Collectively, these prior results make the assumption that L admits a low-rank minimizer more natural.” -- “more naturally”?
* Page 6, line 296, column 1: Missing subscript *, currently reading as a dot product where one of the components is 0.
* Page 11, line 561, Property 3: ∇f(X)UV⊺  -- ∇f(X),UV⊺ {needs comma}
* Page 11, line 571: “regarless” -- “regardless”
* Page 12, line 658: “then, which requires independent reasoning, and then ..” {doesn’t need both then’s}
* Page 17, line 926: Additional epsilon ϵ used in notation that isn’t referenced anywhere else. Currently assuming this is a typo of ε considering the context of the proof.

**Other Strengths And Weaknesses:**

Discussed in other sections. None

**Questions For Authors:**

With the experimentation, is it not true that the use of large variance in initialisation can cause poor performance for alternative reasons? (i.e., initialising with kaiming gives you better results than having a large initialisation in a simple feedforward neural networks)

**Relation To Broader Scientific Literature:**

This work builds on many existing works and confirms existing empirical results in literature as well as 'common knowledge' of LoRa convergence.

**Theoretical Claims:**

I have attempted to review the theoretical correctness of the proofs.  I have validated Proof A.4 and the lemma proofs in appendix B. I was not able to comprehensively verify the results of the main theorem as it is quite dense and complex.
The assumptions of restricted convexity and restricted smoothness appear reasonable and are empirically backed. However, they do break down at larger rank in Table 2.

---

> ### Author Rebuttal · Authors · 2025-04-01
>
> ### **Common Response (Repeated in all responses)**
>
> First of all, we thank the reviewers for their positive and constructive feedback. We are excited to see that the reviewers are appreciative of our theoretical contributions. Below, we address each of the reviewers' comments individually.
>
> ### **Individual Response**
>
> We thank the reviewer for their meticulous and constructive comments, and we are pleased to hear that the reviewer recognizes the novelty of our theoretical contributions.
>
> **Discussion with Table 2**
>
> We appreciate the reviewer’s encouraging remarks on the results of Table 2 and its contrast with over-parameterization. We agree this point is interesting enough to merit further emphasis, so, following the reviewer’s suggestion, we will highlight this discussion more prominently in our revised paper.
>
> **Correction of typos**
>
> We have corrected all typos pointed out by the reviewer. Once again, we sincerely thank the reviewer for their thorough reading of our manuscript and their detailed feedback.
>
> **Question on alternative reasons for the failure case**
>
> We thank the reviewer for raising a thoughtful point. Indeed, bad (large) initializations can lead to poor performance due to issues such as activation function saturation (for tanh activation functions) or exploding gradients. In our setup, this phenomenon indeed occurs if the initialization variance is further increased to $\mathcal{N}(0, \frac{1}{2])$, as demonstrated in the figure in the link below.
>
> https://drive.google.com/file/d/14ui2E-hmg7E7LvaUp2X0hBPx9Zb6AQZp/view?usp=sharing
>
> However, it is unlikely that the behavior observed in Figure 2 of our main text is caused by these issues, as the gradient norm remains stable and nonzero throughout training, as demonstrated in the figure in the link below.
>
> https://drive.google.com/file/d/1w7k0dGcjFvBebFtw8OCqaKZb3BP-1180/view?usp=sharing

---

### Decision · Program_Chairs · 2025-05-01

**Decision:**

Accept (oral)

**Comment:**

**Paper summary**

This paper studies the effect of initialization with less restrictive assumptions. Based on the reviewer's comments, the author theoretically analyzed that a low rank and low magnitude initialization in LoRA models results in convergence towards a global minimum, while larger rank models with larger initialization variance result in convergence towards spurious local minima with high probability.

**Recommendation Justification**

- *Reviewers quality and reviews quality*: Most of the reviewers are familiar with LoRA, either in theory or in more practical terms. It appears that the author was able to convey their message well. The reviews are generally positive with 3,3,3,4 scores in the end. Most of the reviewers also comment on the author's rebuttal and at least acknowledge it. So, I think this evaluation is fair.

I also think this paper is quite interesting in terms of its analysis result. It deserves to be published at ICML this year.